# Internal states drive nutrient homeostasis by modulating exploration-exploitation trade-off

Verónica María Corrales-Carvajal[1,2], Aldo A Faisal[2,3,4], Carlos Ribeiro[1]*

[1]Champalimaud Neuroscience Programme, Champalimaud Centre for the Unknown, Lisbon, Portugal; [2]Department of Bioengineering, Imperial College London, London, United Kingdom; [3]Department of Computing, Imperial College London, London, United Kingdom; [4]Integrative Biology Division, MRC Clinical Sciences Centre, London, United Kingdom

**Abstract** Internal states can profoundly alter the behavior of animals. A quantitative understanding of the behavioral changes upon metabolic challenges is key to a mechanistic dissection of how animals maintain nutritional homeostasis. We used an automated video tracking setup to characterize how amino acid and reproductive states interact to shape exploitation and exploration decisions taken by adult *Drosophila melanogaster*. We find that these two states have specific effects on the decisions to stop at and leave proteinaceous food patches. Furthermore, the internal nutrient state defines the exploration-exploitation trade-off: nutrient-deprived flies focus on specific patches while satiated flies explore more globally. Finally, we show that olfaction mediates the efficient recognition of yeast as an appropriate protein source in mated females and that octopamine is specifically required to mediate homeostatic postmating responses without affecting internal nutrient sensing. Internal states therefore modulate specific aspects of exploitation and exploration to change nutrient selection.

*For correspondence: carlos.ribeiro@neuro.fchampalimaud.org

**Competing interests:** The authors declare that no competing interests exist.

## Introduction

Nutrition is key for optimizing the evolutionary fitness of animals. Accordingly, many organisms are able to select the nutrients that fulfill their current needs. Recent work has highlighted the importance of the balance of dietary carbohydrates and proteins/amino acids (AAs) for overall mortality, fecundity and lifespan in most species (*Fontana and Partridge, 2015*) ranging from *Drosophila* (*Grandison et al., 2009*; *Lee et al., 2008*; *Skorupa et al., 2008*) to rodents (*Solon-Biet et al., 2014, 2015*) and humans (*Levine et al., 2014*). The emerging picture is that there is a trade-off between reproduction and longevity driven by the protein-to-carbohydrate ratio in the diet: a low ratio extends lifespan but reduces reproductive output, while a high ratio reduces lifespan but promotes offspring production (*Simpson et al., 2015*). The mechanisms by which the brain shapes behavioral output during dietary balancing to solve this ethologically relevant trade-off are still largely unknown.

Significant advances have been made in our understanding of the neural circuitry underlying decision-making (*Barron et al., 2015*; *Lisman, 2015*). But we are only beginning to understand how the internal state of an animal dictates the selection of specific actions (*Krashes et al., 2009*; *Sternson, 2013*). This question becomes particularly relevant in value-based decision making, such as nutrient balancing, where the value of the available options is dependent on the current needs of the animal (*Itskov and Ribeiro, 2013*; *Ribeiro and Dickson, 2010*; *Simpson and Raubenheimer, 2012*). Thus, the behavioral strategies animals use to adapt nutrient decisions to their internal states

**eLife digest** When making decisions, animals, including humans, do not always choose the same option. One reason for this is that their "internal state" changes the value of different options. This is particularly evident when deciding what type of food to eat. Depending on which nutrients the animal needs, it will choose to eat different foods.

Amino acids are key nutrients that affect health, lifespan and reproduction. Female fruit flies that have recently mated, for example, eat more amino acids in order to obtain the raw materials required to produce eggs. Despite the importance of amino acids, little was known about how animal behavior changes in response to a lack of this nutrient.

Corrales-Carvajal et al. used a video tracking system to measure the time that fruit flies – some of which had a need for amino acids – spent feeding on patches of yeast (which are rich in amino acids) versus patches of sucrose. Recently mated females – and virgins that had been fed a diet lacking in amino acids – consumed more yeast than sucrose, whereas virgin females that were not amino acid deficient showed the opposite pattern. To bias the fly toward eating the right food for their needs, several aspects of the fly's behavior changed, including the number and length of individual feeding bouts. These different behaviors did not all change at the same time.

The pattern of exploration taken by the flies also depended on their need for amino acids. Amino acid deficient flies spent most of their time near known yeast patches. By contrast, fully fed flies adopted a riskier foraging strategy, moving away from known sources of food to explore their environment more widely. In common with humans, the flies relied upon their sense of smell to efficiently identify different types of food.

Overall, the results presented by Corrales-Carvajal et al. provide us with a detailed understanding about how changes to the internal state of the fly affect its behavior. The next step will be to use the powerful genetic tools available for studying fruit flies to reveal the neural circuits and molecular mechanisms that help animals find the types of food that they need.

provide an ethologically relevant framework to understand how internal states change behavior to mediate value-based decisions.

The fly has emerged as an important model to study complex computational tasks due to the availability of sophisticated genetic tools (*Luo et al., 2008*; *Olsen and Wilson, 2008*), a numerically simple nervous system, and the advent of methods to quantitatively characterize behavior. Advanced computational tools have been applied successfully in *Drosophila* to study for example chemotaxis (*Gomez-Marin et al., 2011*; *van Breugel and Dickinson, 2014*), action mapping (*Berman et al., 2014*), aggression and courtship (*Coen et al., 2016*; *Dankert et al., 2009*), fly-fly interactions (*Branson et al., 2009*; *Schneider et al., 2012*), and predator avoidance (*Muijres et al., 2014*). This recent quantitative approach to behavioral analysis has given rise to the field of computational ethology: the use of computerized tools to measure behavior automatically, to characterize and describe it quantitatively, and to explore patterns which can explain the principles governing it (*Anderson and Perona, 2014*). When combined with powerful genetic approaches (*Bath et al., 2014*; *Ohyama et al., 2015*) the fine description of behavior afforded by these methods will allow us to make significant steps forward in our understanding of the neuronal circuits and molecular pathways that mediate behavior.

Flies can detect and behaviorally compensate for the lack or imbalance of proteins and amino acids in the food (*Bjordal et al., 2014*; *Ribeiro and Dickson, 2010*; *Vargas et al., 2010*) and adapt their salt and protein intake to their current mating state (*Walker et al., 2015*). The current nutrient state is thought to be read out directly by the nervous system through the action of nutrient-sensitive mechanisms such as the TOR and GCN2 pathways (*Bjordal et al., 2014*; *Chantranupong et al., 2015*; *Ribeiro and Dickson, 2010*). Mating acts on salt and yeast appetite via the action of male-derived Sex Peptide acting on the Sex Peptide Receptor in female reproductive tract neurons, and the resultant silencing of downstream SAG neurons (*Feng et al., 2014*; *Ribeiro and Dickson, 2010*; *Walker et al., 2015*). SAG neurons have been proposed to then change chemosensory processing to modify nutrient intake (*Walker et al., 2015*). The recent development of technologies that can

measure the flies' feeding behavior quantitatively (*Itskov et al., 2014*; *Ro et al., 2014*; *Yapici et al., 2016*) gives access to the fine structure of the feeding program, and how flies homeostatically modulate this program according to their internal state. However, the further structure of foraging decisions, such as arriving at or leaving a specific food patch, and how flies balance the trade-off between exploiting a needed nutrient resource and exploring the surrounding environment to discover new resources, is still poorly understood. Understanding how internal states change the behavioral strategies of an animal should allow us to understand how the animal manages to maintain nutrient homeostasis.

Here, we developed a quantitative value-based decision making paradigm to study the foraging strategies implemented by adult *Drosophila melanogaster* to reach protein homeostasis. We use an automated video tracking setup to characterize the exploitation and exploration of sucrose and yeast patches by flies in different dietary amino acid and mating states. We found that metabolic state and mating modulate the decisions to stop at a yeast patch and leave it. Furthermore, we describe how the internal deficit of dietary amino acids increases exploitation of proteinaceous patches and restricts global exploration and how these behaviors dynamically shift towards increasing exploration as the fly reaches satiation. Importantly, we provide two examples on how our paradigm can be used in the dissection of the genetic and neuronal mechanisms underlying nutrient decisions: First, we show that olfaction is not required to reach protein homeostasis, but that it mediates the efficient recognition of yeast as an appropriate food source in mated females. Second, we show that octopamine mediates homeostatic postmating responses, but not the effects of internal sensing of amino acid deprivation state. Our study provides a quantitative description of how the fly changes behavioral decisions to achieve homeostatic nutrient balancing as well as initial insights into the mechanisms underlying protein homeostasis.

## Results

### Automated monitoring of nutrient choices using image-based tracking

Animals are able to adapt their feeding preference towards a particular food in response to their current needs (*Dethier, 1976*; *Griffioen-Roose et al., 2012*; *Itskov and Ribeiro, 2013*; *Warwick et al., 2009*). However, the behavioral strategies used by animals to make feeding decisions according to their internal state are currently largely unknown. To capture how flies decide what food to eat, we built an automated image-based tracking setup (*Figure 1A*) that captures the position of a single *Drosophila melanogaster* in a foraging arena (*Figure 1B*) containing 9 yeast patches (amino acid source) and 9 sucrose patches (carbohydrate source) of equal concentration.

The distribution of the food patches was designed to promote frequent encounters with food sources, such that nutritional decisions, rather than food finding, determine the fly's food exploitation strategies. We recorded the behavior of the fly over two hours during these nutritional decisions, and developed custom software to track the position of the fly's body and head centroids (all tracking data generated in this study are available for download from the Dryad repository [*Corrales-Carvajal et al., 2016*]). We then extracted multiple kinematic parameters (see Materials and methods for detailed list) and computed the locomotor activity and the distance of the fly from each food patch during the whole duration of the assay (*Figure 1C and D* and *Video 1*). Upon a detailed analysis of the distribution of head speeds when the flies were inside or outside food patches (*Figure 1—figure supplement 1A*) we decided to use two speed thresholds to split the locomotor activity of the flies into three types: *resting* (speed $\leq$ 0.2 mm/s), *micromovement* (0.2 mm/s < speed $\leq$ 2 mm/s) and *walking* (speed > 2 mm/s). Furthermore, slow walking bouts (2 mm/s < speed < 4 mm/s) that were coupled with a rapid change in angular speed were defined as *sharp turns* (2 mm/s < speed < 4 mm/s and |angular speed| $\geq$ 125°/s) (*Figure 1C and D*).

To characterize the behaviors that occur during these defined locomotor activity types, we manually annotated resting, feeding, grooming and walking events and assigned them to the corresponding speed profiles. In agreement with previous studies (*Martin, 2004*; *Robie et al., 2010*; *Zou et al., 2011*), we found that more than 80% of the speeds displayed during manually annotated resting or walking periods were below 0.2 mm/s or above 2 mm/s, respectively (*Figure 1—figure supplement 1B*). Furthermore, we reasoned that micromovements could correspond to either grooming or feeding. Indeed, 70% of grooming fell in the micromovement category; while for manually-annotated

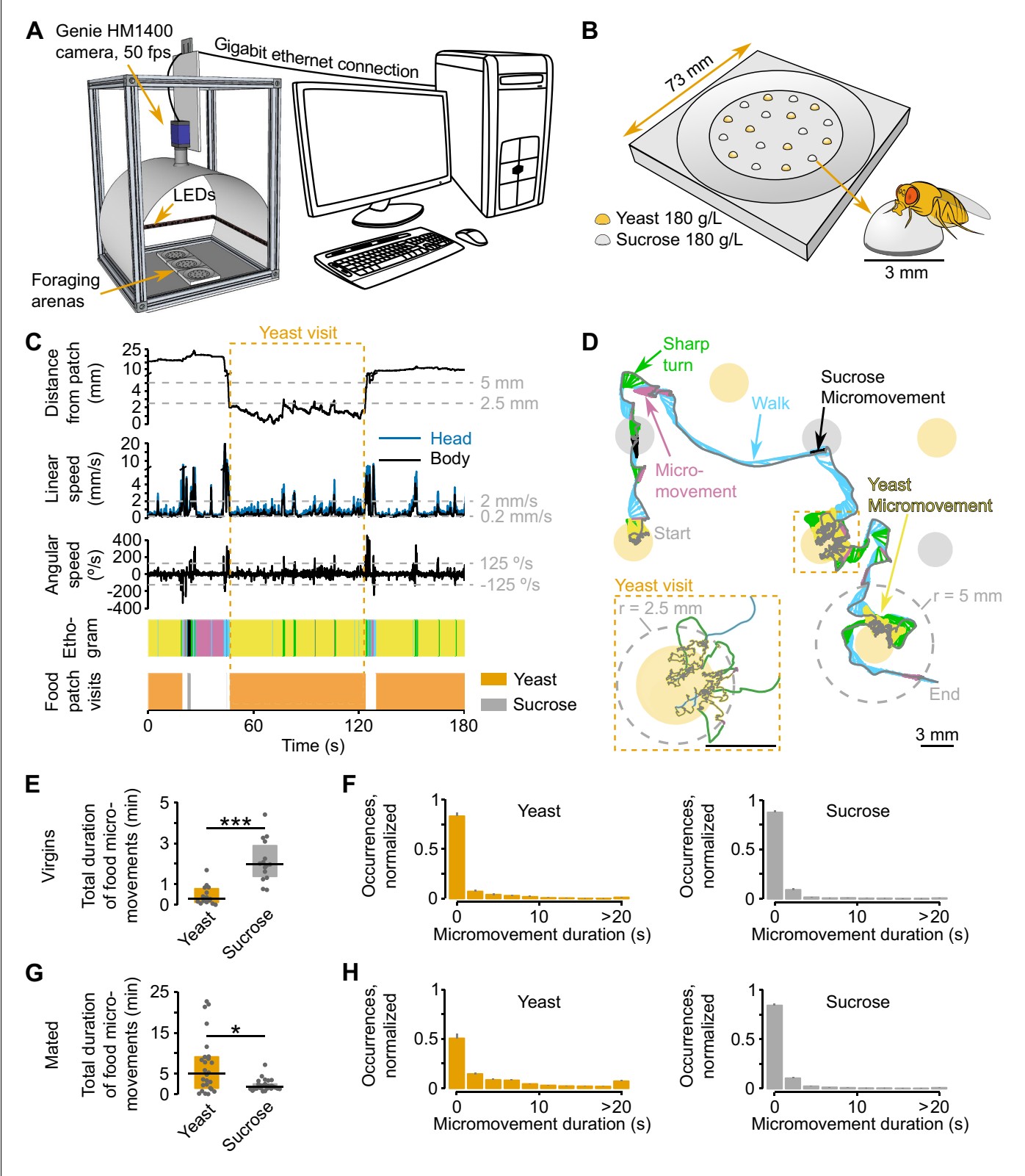

**Figure 1.** Automated monitoring of nutrient choices using image-based tracking. (A) Schematic of the image-based tracking setup. (B) Schematic of the foraging arena, containing an inner flat circular area with 9 sucrose (carbohydrate source) and 9 yeast (amino acid source) patches. All patches had a concentration of 180 g/L of the corresponding substrate. Each food patch has an approximate diameter of 3 mm which is approximately the body length of the experimental flies. (C) Example of the kinematic parameters and behavior classification associated to the representative trajectory shown

*Figure 1 continued on next page*

*Figure 1 continued*

in (**D**). Dashed gray horizontal lines indicate the thresholds used for behavior classification, definition of yeast and sucrose micromovements and food patch visits (see materials and methods). Dashed orange rectangle marks the beginning and end of the yeast visit (see inset in **D**). The different colors in the ethogram correspond to the behaviors labeled with the same color in (**D**). (**D**) Representative trajectory of a fly walking in the arena. Filled circles represent food patches. Gray and colored trajectories correspond to head and body centroid position, respectively. Small arrows in between both trajectories indicate body orientation. The color code for the different behaviors is indicated by the colored labels. Inset: a yeast visit is defined as a group of consecutive yeast micromovements, in which the head distance to the center of the food patch was never >5 mm (gray dashed line in the main trajectory). (**E,G**) Total duration of yeast and sucrose micromovements for virgin, n = 15 (**E**) and mated, n = 26 (**G**) female flies fed with the AA+ rich diet. (**F,H**) Distribution of yeast and sucrose micromovement durations for virgin (**F**) and mated (**H**) female flies fed with the rich diet. Bin size: 2.2 s. *p<0.05, ***p<0.001, significance was tested by Wilcoxon rank-sum test. In panels **E** and **G** and in the following figures in which boxplots are used, the black line represents the median, colored boxes represent inter-quartile range (IQR) and gray dots represent the value of the y-axis parameter for single flies.

The following figure supplement is available for figure 1:

**Figure supplement 1.** Ground-truthing of behavior.

feeding bouts, half of these periods were categorized as micromovements, the other half occurred at low speeds and were thus classified as resting. However, flies showed a very low rate of proboscis extension during feeding bouts at <0.2 mm/s (data not shown) and we therefore reasoned that these slow bouts had little contribution to the amount of food ingested. For this reason, we decided to use the time the fly was performing micromovements when its head was in contact with the food patch as a proxy for the time the fly spent feeding (henceforth termed *yeast micromovements* or *sucrose micromovements*). To strengthen the argument that these micromovement periods within a food patch represented mostly feeding bouts and not grooming, we used the annotated video segments to quantify the percentage of feeding and grooming during a food micromovement bout. Indeed, we observed that 92.2% of the yeast micromovements and 70.6% of the sucrose micromovements corresponded to feeding bouts (*Figure 1—figure supplement 1C*). Hence *sucrose* and *yeast micromovements* are a good way to capture the periods the fly spends feeding on a food patch.

To start exploring how flies with different internal states react to the different foods, we used this metric to characterize the behavior of virgin and mated females that were previously fed a rich diet. Virgin flies displayed a preference for sucrose over yeast over the total time of the assay, while the opposite was observed in mated females (*Figure 1E and G*). A closer look at the duration of micromovements on the two food sources, revealed very similar duration profiles between yeast and sucrose for virgin females, while a higher prevalence of long events (≥20 s) on yeast when compared to sucrose was observed in mated flies (*Figure 1F and H*). These results suggest that for mated females, yeast has a higher salience as food source, even in fully-fed conditions. These observations are in accord with previous reports showing that mating leads to a switch in yeast preference in flies (*Ribeiro and Dickson, 2010*; *Vargas et al., 2010*; *Walker et al., 2015*). Thus, the analysis of food micromovements allows us to capture previously-described changes in food preference elicited by mating. Furthermore, these results demonstrate that one way in which mating increases yeast preference is by inducing long feeding bouts, allowing us to make first conclusions about the behavioral mechanisms behind changes in food choice.

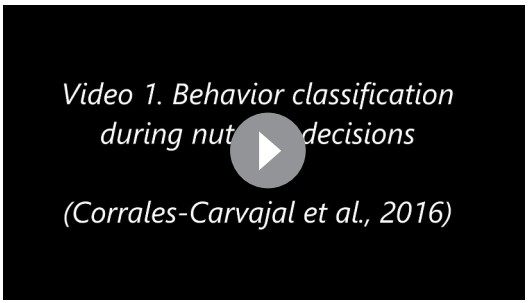

*Video 1. Behavior classification during nutrient decisions*

*(Corrales-Carvajal et al., 2016)*

**Video 1.** Behavior classification during nutrient decisions. A 20-s-segment of the trajectory depicted in *Figure 1C–D*, starting on second 40 and following the same color code. The first 7 s of the video are slowed-down 0.5 x, as indicated by the white label at the top right corner of the video frame with the fly.

## Flies increase yeast feeding and micromovements in response to amino acid challenges and mating

A key question in nutritional neuroscience is how animals homeostatically compensate for the lack of specific nutrients (*Dethier, 1976*; *Itskov and Ribeiro, 2013*; *Simpson and Raubenheimer, 2012*). A concrete example of this homeostatic regulation of feeding behavior is the robust increase in preference for yeast when flies are deprived of proteinaceous food (*Ribeiro and Dickson, 2010*; *Vargas et al., 2010*). To study the behavioral strategies underlying nutritional homeostasis, we manipulated the metabolic state of the flies by letting them feed *ad libitum* on a chemically defined (holidic) medium (*Piper et al., 2013*) during three days prior to the foraging assay. This holidic medium allows us to specifically manipulate amino acids (AA) in the diet, leaving the other macronutrients and micronutrients intact. Previous work has identified three different AA compositions having different impacts on reproduction in mated females: *AA+ rich* (supporting a high rate of egg laying), *AA+ suboptimal* (supporting a lower rate of egg laying) and *AA-* (leading to a dramatic reduction in egg laying) (*Piper et al., 2013*; *Figure 2A*). Furthermore, to better understand how internal metabolic state and mating state interact at the behavioral level we also analyzed virgin females pre-fed these different diets.

To quantify the microstructure of the feeding behavior of flies with different internal states, we used the flyPAD technology (*Figure 2—figure supplement 1A*), which allowed us to decompose the feeding motor pattern into 'sips' (*Itskov et al., 2014*). As the number of sips correlates strongly with food intake, this method enabled us to precisely measure the impact of internal states on feeding decisions. Consistent with previous observations (*Ribeiro and Dickson, 2010*; *Vargas et al., 2010*; *Walker et al., 2015*) (*Figure 1E*), virgin flies showed very little interest in yeast during the whole assay, as measured by the total number of yeast sips (*Figure 2Bi*). Yeast feeding increased with AA deprivation (*Figure 2Bii*), and mating (*Figure 2Biii*). Notably, AA-challenged mated females showed a strong increase in the number of yeast sips (*Figure 2Biv and v*) with the highest rate of yeast feeding in mated flies completely deprived of AAs (*Figure 2Bv*). We next asked whether these differences in feeding behavior could be captured using the yeast and sucrose micromovements measured using the tracking setup. Indeed, we observed that the yeast micromovements increased in the same way as the yeast sips after AA challenges in virgin and mated females (*Figure 2C*).

Importantly none of these internal state changes led to an increase in the total number of sucrose sips (*Figure 2—figure supplement 1B*) or in the total duration of sucrose micromovements (*Figure 2—figure supplement 2*), highlighting the dietary specificity of the manipulation and allowing us to focus our subsequent analysis on the fly's behavior towards yeast patches. Flies are therefore capable of sensing deficits in AAs and of compensating by specifically increasing feeding and micromovements on yeast, an AA-rich substrate. Furthermore, this homeostatic response is modulated by the mating state of the fly. Our tracking approach is therefore now a validated strategy to uncover the changes in behavioral strategies elicited by different internal states and how these changes allow the animal to reach homeostasis.

## Flies show high inter-individual variability in the response to yeast

We investigated the dynamics of yeast micromovements by comparing the ethogram of each individual fly along the two hours of the assay and across all the internal state conditions tested (*Figure 2D*, yeast micromovements are shown in yellow). This type of visualization revealed that the behavior towards yeast was highly variable. The observed increase in the total duration of yeast micromovements across the different internal state conditions seems to come from the combination of two factors: on one hand, the proportion of flies that showed any interest in yeast at all (*Figure 2—figure supplement 3A*) and on the other hand, the strength of the interest displayed by these flies, measured by the total duration of yeast micromovements. The behavior towards yeast was also highly variable across individuals of the same condition. For example, the total duration of yeast micromovements displayed by AA-deprived flies ranged from 5 to 59 min. Still, the initial steep increase in yeast micromovements during the first 30 min of the assay was very consistent (*Figure 2Ev* and *Figure 2—figure supplement 1C*). Overall, the variability increased as a function of deprivation (*Figure 2—figure supplement 3B*). The reaction to internal state changes is therefore highly variable across individuals. However, full AA deprivation in mated females leads to a robust population-wide effect, highlighting the importance of AAs for the animal.

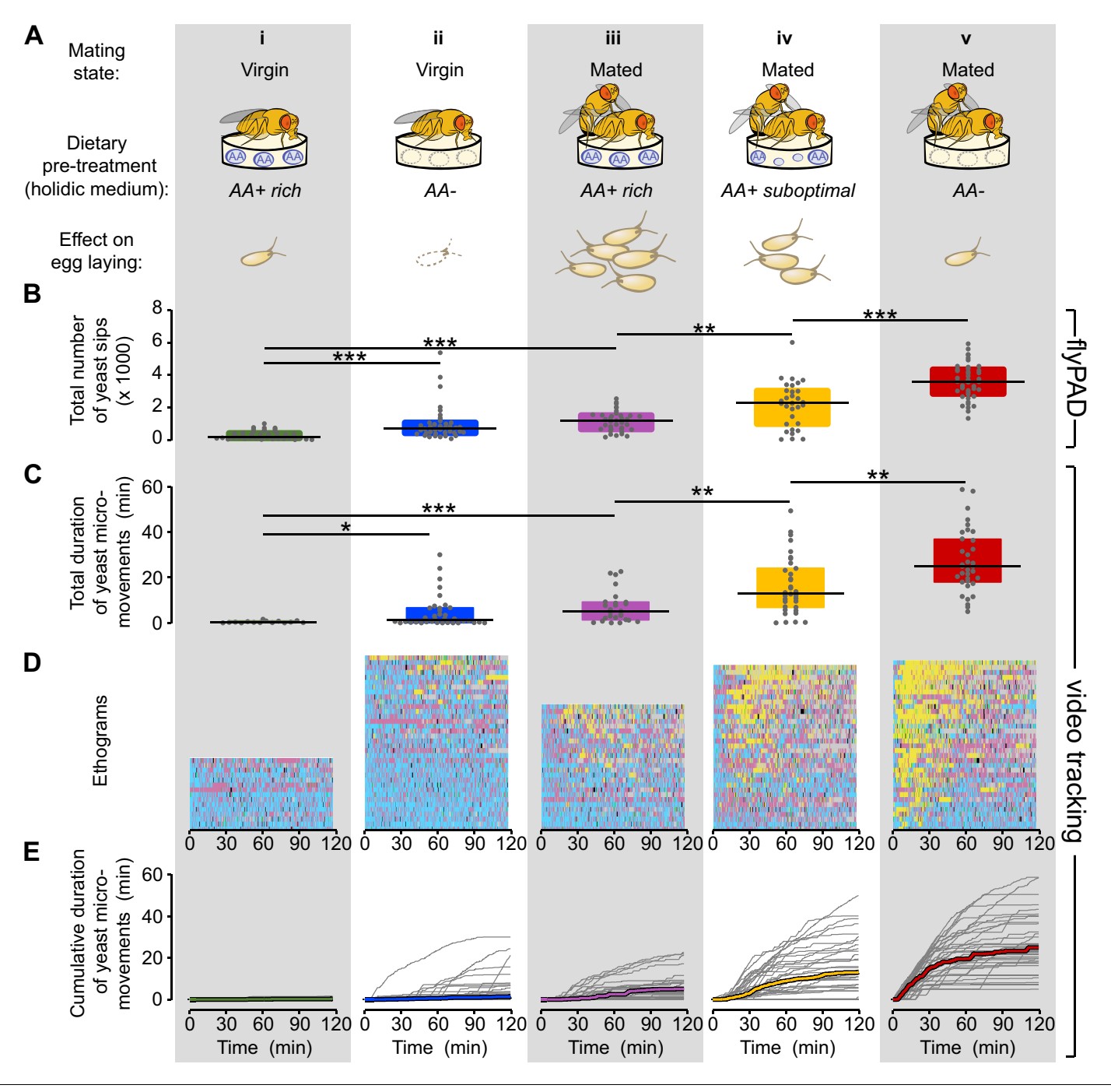

**Figure 2.** Flies increase yeast feeding and micromovements in response to amino acid challenges and mating. (A) Graphical representation of the five internal states tested and the resulting reproductive output as reported by *Piper et al. (2013)*, all flies were pre-fed during three days with the indicated holidic medium: (i) Virgin AA+ rich, (ii) Virgin AA+, (iii) Mated AA+ rich, (iv) Mated AA+ suboptimal, (v) Mated AA−. (B) Effect of internal states on the total number of yeast sips obtained using flyPAD assay (n = 32–43). (C) Effect of internal states on the total duration of yeast micromovements obtained from the video tracking assay (n = 15–35). (D) Behaviors displayed by single flies in the five internal states indicated in (A), during the video tracking assay. Each row represents the ethogram of a single fly, following the same color code used in *Figure 1D*. Yellow: yeast micromovements. Black: sucrose micromovements. Pink: micromovements outside the food patches. Blue: walking bouts. Gray: resting bouts. Green: sharp turns. (E) Dynamics of yeast micromovements quantified as the cumulative duration of yeast micromovements. Gray lines correspond to single flies. Thick colored lines indicate median. *p<0.05, **p<0.01, ***p<0.001, significance was tested by Wilcoxon rank-sum test with Bonferroni correction.

The following figure supplements are available for figure 2:

*Figure 2 continued on next page*

*Figure 2 continued*

**Figure supplement 1.** flyPAD setup, sucrose sips and yeast sips dynamics.

**Figure supplement 2.** Sucrose micromovements.

**Figure supplement 3.** Fraction of yeast non-eaters and coefficient of variation for yeast micromovements.

## Metabolic state and mating modulate the probability of stopping at a yeast patch and leaving it

To feed, flies need to stay on food patches. We decided to call these events *visits* (*Figure 1C* and inset in *Figure 1D*). A *visit* is defined as all consecutive bouts of micromovements on the same patch, for as long as the fly stayed in close proximity of the patch. As we observed in the total duration of yeast micromovements, the total duration of yeast visits increased as a result of mating and AA deprivation (*Figure 3A*). Therefore, we hypothesized that the fly increases yeast intake by changing different aspects of its foraging decisions, such as *approaching* a patch more often, *stopping at* it more and/or *leaving* it less often. We measured approaching, stopping and leaving decisions by quantifying the number of encounters, the fraction of encounters in which the fly stops on a patch (*visits*) and the duration of visits, respectively. One easy way to increase the total time on yeast would be to approach yeast patches more often. However, none of the internal state modifications leading to an increase in yeast intake caused an increase in the total number of yeast encounters (*Figure 3—figure supplement 1A*). Furthermore, the rate of encounters remains constant across internal states, with the exception of the mated fully AA-deprived females (*Figure 3B*), which had a low absolute number of encounters (*Figure 3—figure supplement 1A*). To explain the behavioral changes underlying homeostasis, we focused on the decision to stop at a yeast patch (*Figure 3C*) and leave it (*Figure 3D*).

We found that in virgins, AA deprivation had a specific effect as it only modulated the decision to leave a patch, with deprived virgins showing longer visits (*Figure 3D* and *Figure 3—figure supplement 1B*). Mating also modulated the decision to leave, as fully-fed mated females took longer to leave a yeast patch than virgins (*Figure 3D* and *Figure 3—figure supplement 1B*), and, to a smaller degree, had a higher probability of stopping at a proteinaceous food patch upon encounter (*Figure 3C*). Surprisingly, pre-feeding mated flies with the suboptimal diet caused a dramatic increase in the probability of stopping at a yeast patch (*Figure 3C*). The strong effect on the decision to stop shows that flies are able to homeostatically modify their behavior in response to even subtle dietary differences that have a negative impact on their fitness (*Piper et al., 2013*). This is even more striking considering that the removal of all AAs does not lead to further changes in the *stopping* and *leaving* decisions, despite its drastic impact on egg production and yeast feeding (*Figure 2*).

We showed above that there is considerable variability across individuals in their behavioral response towards yeast. This was also the case for the strategies each mated fly chose to compensate for both AA challenges. We observed that these flies reached the same total times on yeast by mixing strategies in different ways: some flies had fewer but longer visits, while others had a higher number of visits, but each visit was shorter (*Figure 3E*). Taken together, these data show that both metabolic and mating states significantly change the decisions to stop at a yeast patch and leave it. Furthermore, the strongest effect is observed when both states act together, as seen in AA-challenged mated females. The specific behavioral strategy each fly employs to reach homeostasis, however, varies widely.

## The lack of dietary AAs increases exploitation and local exploration of yeast patches

We have shown that AA deprivation leads to a 1.6-fold increase in yeast feeding when compared to the suboptimal diet treatment (*Figure 2B*). Surprisingly, however, a change of this magnitude is not visible in the total duration of the yeast visits (*Figure 3A*), nor is this homeostatic effect recapitulated in changes in specific foraging decisions (*Figure 3*). We therefore speculated that instead of

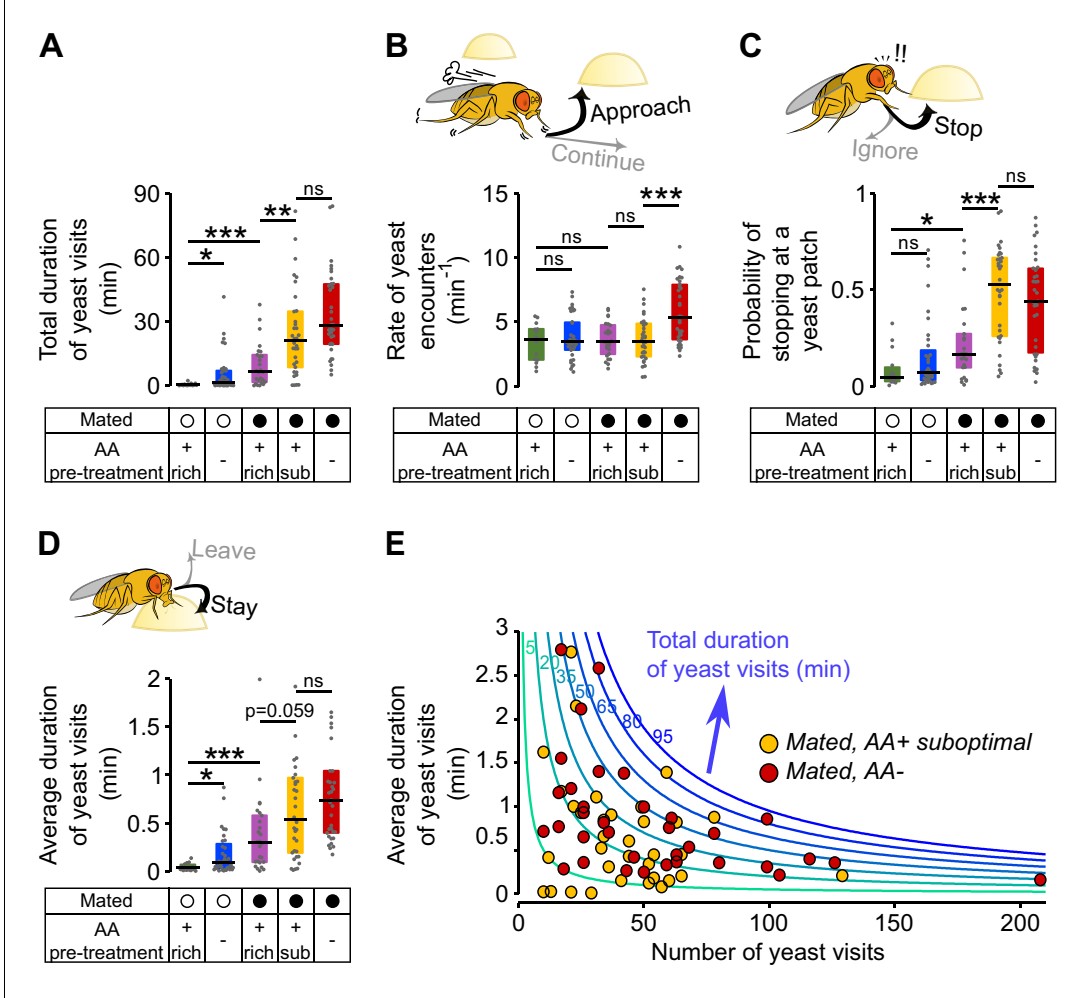

**Figure 3.** Metabolic state and mating modulate the probability of stopping at a yeast patch and leaving it. (A) Effect of internal states on the total duration of yeast visits. Experimental groups are the ones shown in *Figure 2*: Mated (filled circles) and virgin (open circles) females pre-fed three types of holidic media: *AA+ rich, AA+ suboptimal* and *AA−*. (B) Effect of internal states on the decision to approach a yeast patch quantified as the number of yeast encounters per minute of walking outside the food patches (rate of yeast encounters). (C) Effect of internal states on the decision to stop at a yeast patch quantified as the fraction of yeast encounters in which the fly stopped at the yeast patch. (D) Effect of internal states in the decision to leave a yeast patch quantified as the average duration of yeast visits. (E) Combination of foraging strategies (total number of visits in x-axis and average duration of those visits in y-axis) to reach different total durations of yeast visits (green to blue lines), for individual AA-challenged mated flies: pre-fed either a suboptimal diet (yellow circles) or an AA- diet (red circles). *ns*, not significant (p≥0.05), *p<0.05, **p<0.01, ***p<0.001, significance was tested by Wilcoxon rank-sum test with Bonferroni correction.

The following figure supplement is available for figure 3:

**Figure supplement 1.** Yeast encounters and probability of leaving.

modulating exploratory decisions, a lack of AAs could increase the motivation of the flies to exploit the yeast patch. Indeed, the time course of yeast visits (*Figure 4A* and *Figure 4—figure supplement 1A*) shows that AA-deprived flies displayed a sharp increase in the total duration of yeast visits during the first minutes, while flies pre-fed a suboptimal AA diet displayed a much more delayed and shallower increase in this parameter. As these early visits were also longer (*Figure 4—figure supplement 1B*), we measured the time it took each fly to engage in its first 'long' (≥30 s) visit (*Figure 4—figure supplement 1C and D*), and found that AA-deprived flies indeed attained their first long yeast visit much sooner than flies fed a suboptimal diet (*Figure 4B*): the median latency for AA-deprived flies was just 4.38 min (IQR = 2.08–7.7), which was three times shorter than the 12.37 min

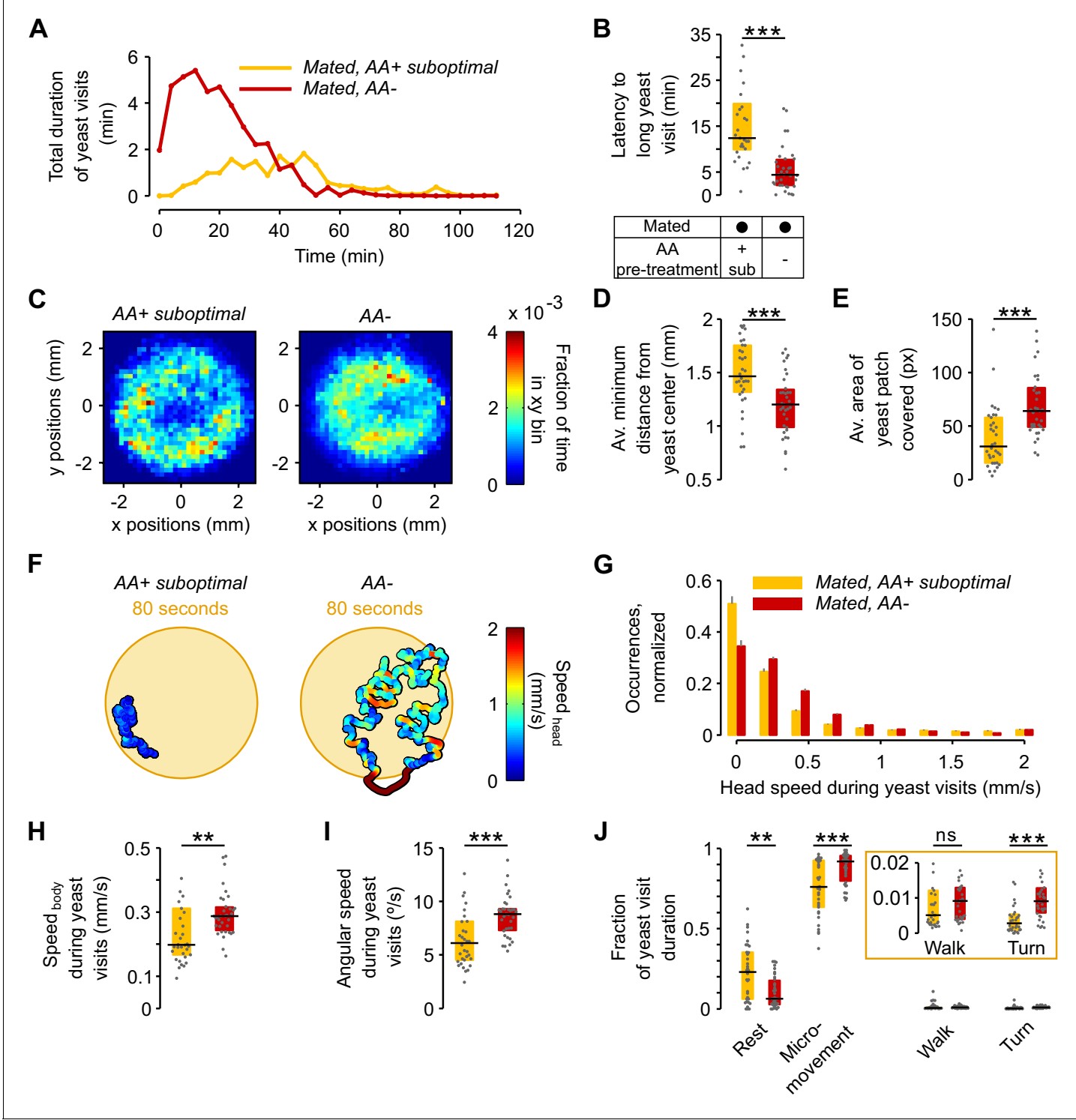

**Figure 4.** The lack of dietary AAs increases exploitation and local exploration of yeast patches. (A) Rolling median of the total duration of yeast visits using a 5 min window and a step of 4 min for flies pre-fed a suboptimal diet (yellow) or AA− diet (red). (B) Effect of AA deprivation on the time elapsed until the fly engages in the first 'long' (≥30 s) yeast visit. (C) Histogram of the x-y relative position of all mated flies pre-fed a suboptimal diet (left) or a AA− diet (right) with respect to the center of the yeast patch (0,0). The pixel color indicates the fraction of time that flies in the indicated condition spent in the corresponding location bin. (D) Effect of AA deprivation on the average minimum distance to the center of the yeast patch, during a yeast visit. (E) Effect of AA deprivation on the average area covered during a yeast visit. (F) Example trajectories of head position during a yeast visit for a fly of the indicated condition. Hot colors indicate higher head speeds. (G–J) Effect of AA deprivation on the locomotor activity of mated flies during yeast

*Figure 4 continued on next page*

*Figure 4 continued*

visits: (G) average histogram of head speeds, (H) body centroid speed, (I) angular speed and (J) proportion of the indicted behaviors during yeast visits. *ns*, not significant (p≥0.05), **p<0.01, ***p<0.001, significance was tested by Wilcoxon rank-sum test with Bonferroni correction. Panels B, D, E, H–J compare the indicated parameters between mated flies pre-fed a suboptimal diet (yellow) and mated flies pre-fed an AA− diet (red).

The following figure supplements are available for figure 4:

**Figure supplement 1.** Yeast visits dynamics and latency.

**Figure supplement 2.** No effect in local exploration of yeast patches for flies pre-fed a suboptimal diet.

**Figure supplement 3.** Modulation of yeast feeding program microstructure by AA challenges.

(IQR = 19.87–9.86) observed in flies fed with the suboptimal diet. These results therefore suggest that AA-deprived flies are indeed more motivated to exploit yeast patches.

We next asked if AA deprivation could also induce differences in the way flies behaved on the yeast patches. When we plotted the distribution of the positions of the flies on the proteinaceous food patches, we observed that AA-deprived flies covered the patches more homogeneously than flies kept on a suboptimal diet, which preferred to stay at the edge of the patch *Figure 4C*). In fact, deprived flies ventured much more into the food patch as quantified by the fact that during a visit, their average minimum distance from the patch center was much smaller (*Figure 4D*) and that they covered a larger area of the resource (*Figure 4E*). These data suggest that AA-deprived flies are not only more motivated to start exploiting a yeast patch but are also more active while on the food patch.

This was further supported when we quantified locomotor activity during each visit to yeast. As visible in the two example trajectories displayed in *Figure 4F*, we observed that deprived flies were more active, displaying higher linear (*Figure 4G and H*) and angular speeds (*Figure 4I*). Accordingly, these flies had fewer resting bouts and more sharp turns (*Figure 4J*). These changes in behavior observed on the food patch were only induced by a complete lack of AAs, as there was no difference in these parameters between mated females pre-fed the rich diet versus those pre-fed the suboptimal diet (*Figure 4—figure supplement 2*). All these data are in agreement with an increase in yeast exploitation upon full AA deprivation. Flies lacking AAs would be more 'eager' to exploit and therefore ingest yeast, leading to a strong increase in yeast feeding as observed using the flyPAD (*Figure 2B*).

Animals homeostatically increase food intake upon food deprivation, by changing the micro-structure of their feeding motor pattern (*Davis and Smith, 1992*; *Itskov et al., 2014*) (*Figure 4—figure supplement 3A*). As video tracking does not give us access to the fine structure of the proboscis motor program, we used the flyPAD technology to characterize the changes in the microstructure of feeding upon AA deprivation. Pre-feeding flies a suboptimal diet led to a decrease only in the inter-burst-interval (IBI) when compared to flies kept on a rich diet (*Figure 4—figure supplement 3B*) while the number of sips in each feeding burst did not change (*Figure 4—figure supplement 3C*). Full AA deprivation, however, led to a strong increase in the number of sips per burst with only a mild further decrease in the IBI. These effects are very similar to those observed upon mating in yeast-deprived females, which leads to both a decrease in the inter-burst interval and an increase in the number of yeast sips per burst (*Walker et al., 2015*).

The volume ingested during a feeding bout is a product of the duration of that bout and the feeding rate. Therefore, we analyzed the rhythmic feeding motor pattern and observed that it was only slightly modified by dietary AA levels (*Figure 4—figure supplement 3D and E*). The mode of the inter-sip-interval distributions decreased from 0.08 s in mated females pre-fed the rich diet to 0.07 s when pre-fed the suboptimal diet (p=0.0045, Wilcoxon rank-sum test with Bonferroni correction), while no further change was observed when they were pre-fed the AA− diet (0.07 s, p=1). However, the mode of the sip duration distributions did not change when mated flies pre-fed a suboptimal diet were compared to females kept on a rich diet (0.12 s, p=0.1196), but it decreased when flies were pre-fed the AA− diet (0.11 s, p=0.0067). Taken together, while AA deprivation has minimal effects on the decision to stop at a proteinaceous food patch and leave it, this metabolic

manipulation leads to drastic changes in its exploitation. The described changes in activity are likely to support an increased intake of the yeast resource, which is further promoted by a change in the feeding motor pattern of the fly. The increase in exploitation can also be interpreted as an increase in local, resource-directed exploration which could aid the micro-optimization of food intake within non-homogenous natural food patches.

## Amino acid challenges reduce global exploration and increase revisits to the same yeast patch

The data presented above clearly demonstrate that different internal states interact to modulate food exploitation. But what could be the effects of internal states on the exploratory behaviors of flies? In order to capture how far the fly would forage to reach the next yeast patch, we calculated three types of transition probabilities: transitions to the same yeast patch, transitions to adjacent yeast patches, and transitions to distant yeast patches. We found that mated flies fed the rich diet had a high probability of transitioning to distant yeast patches (75%) (*Figure 5A and D*), and a lower probability of transitioning to adjacent food patches (25%) (*Figure 5A and E*).

Strikingly, these flies almost never returned to the yeast patch they had just visited (*Figure 5A and F*). Fully-fed flies therefore display a high rate of global exploratory activity, traveling larger distances during their transitions (*Figure 5G*) and mainly choosing to visit distant food patches (as in the example trace). Challenging flies with a suboptimal diet (*Figure 5B*) or one lacking all AAs (*Figure 5C*) significantly altered their exploratory behavior: they strongly reduced their probability of transitioning to distant yeast patches (*Figure 5D*) and increased the probability of transitioning to adjacent yeast patches (*Figure 5E*). Further, in contrast to the fully-fed flies, AA-challenged flies showed a strong increase in their probability of returning to the same yeast patch (*Figure 5F* and *Figure 5—figure supplement 1*). As one would expect, these changes in behavior are also seen as a decrease in the average distance traveled by animals during transitions to yeast (*Figure 5G*). Dietary AA challenges therefore lead to a switch from global to local exploration (see example traces in *Figure 5A–C*). One of the most striking changes is the strong increase in returns to the same yeast patch upon AA deprivation. This change in exploratory strategy leads to an effective additional increase in the time on the same yeast patch without requiring a change in the decision to leave it. Taken together, these changes in exploratory strategy should enable the fly to efficiently increase the intake of yeast while minimizing the distance traveled to the next patch. It also allows the fly to focus on a resource whose quality she knows while avoiding testing food patches of unknown qualities, thereby reducing exploratory risk.

## Flies dynamically adapt their exploitatory and exploratory behavior to their internal AA state

If yeast exploitation and exploration are indeed regulated by the internal AA state of the fly, we hypothesized that flies should dynamically adapt their behavior as their internal state changes over the course of the assay due to satiation. To capture this effect independently from the varying yeast intake dynamics displayed by each fly, we divided the total duration of yeast micromovements of each fly into four periods, which we called 'yeast quartiles' (*Figure 6A*). Each *yeast quartile* consists of 25% of the time that the fly spent in yeast micromovements, but covers a different amount of absolute time in the assay for each fly.

As hypothesized, the flies displayed clear differences in their foraging behavior across the four analyzed quartiles. The effect on exploration was clearly visible in the raw tracking traces for the four quartiles (*Figure 6B*). As the time spent on yeast increased, the average distance traveled to the next yeast patch (*Figure 6C*) and the probability of visiting a distant yeast patch increased (*Figure 6D*), while the probability of revisiting the same yeast patch decreased (*Figure 6E*). Accordingly, parameters related to patch exploitation such as the average minimum distance from the center of the patch (*Figure 6F*), the angular speed on the yeast patch (*Figure 6G*), and the average duration of the yeast visit (*Figure 6H*) reverted to the values observed in fully-fed females (*Figure 6—figure supplement 1A–C*). These results show that flies are capable of dynamically adapting their behavioral strategies according to their current internal state and strengthen the notion that foraging strategies are modified by the AA state of the animal to homeostatically balance the intake of AA-rich foods.

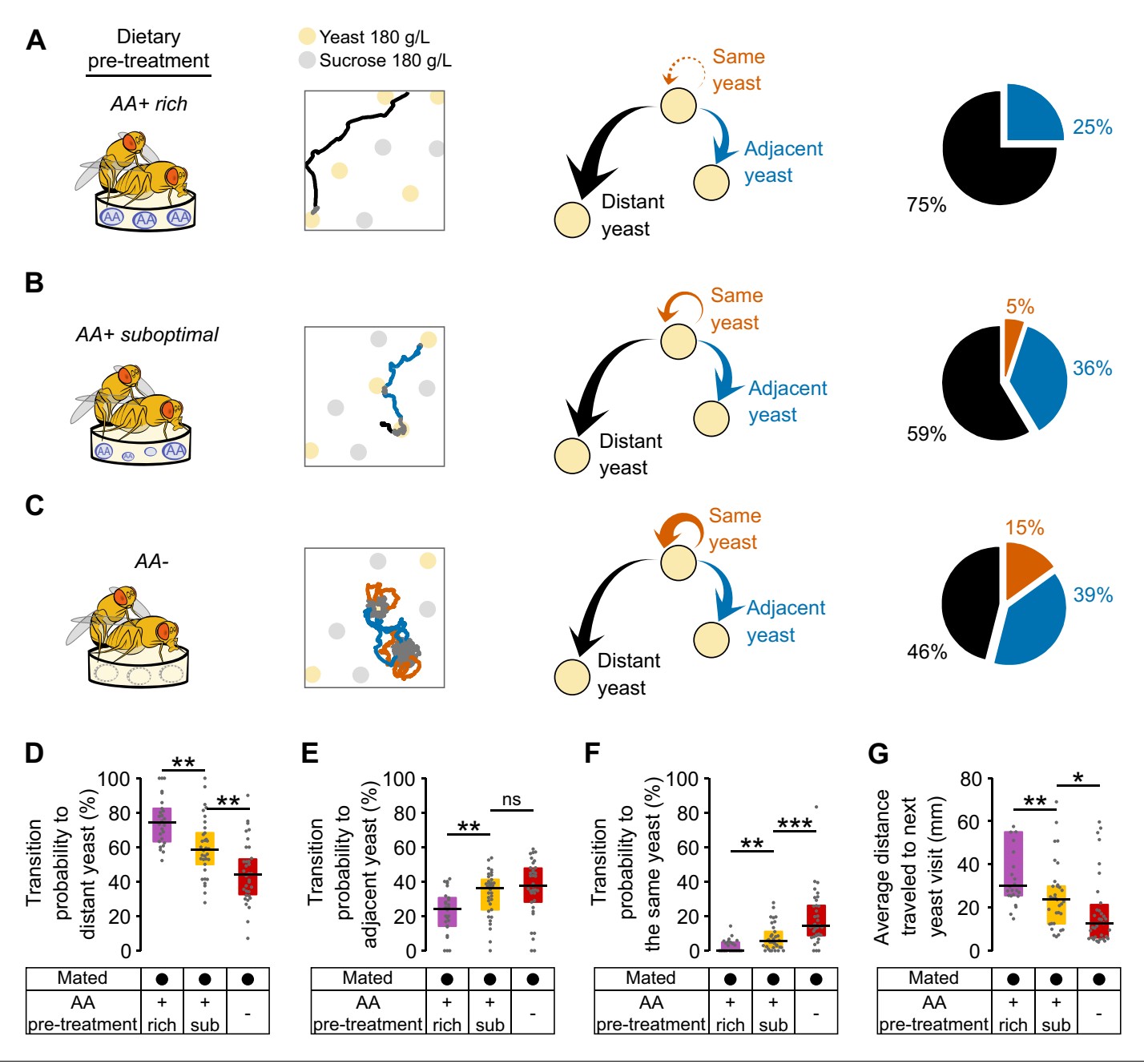

**Figure 5.** Amino acid challenges reduce global exploration and increases revisits to same yeast patch. (A–C) Effect of internal states on exploratory behavior of mated females pre-fed with an AA rich diet (A), an AA suboptimal diet (B) or an AA− diet (C). Example trajectories show head position during a yeast-yeast transition. Arrows and pie charts indicate the transition probabilities to visit three types of yeast patches: the same (orange), an adjacent one (blue) or a distant one (black). (D–F) Comparison of the transition probabilities described in (A–C) across the different diet treatments in mated females. (G) Average distance covered during transitions to yeast visits. *ns*, not significant (p≥0.05), *p<0.05, **p<0.01, ***p<0.001, significance was tested by Wilcoxon rank-sum test with Bonferroni correction.

The following figure supplement is available for figure 5:

**Figure supplement 1.** Dynamics of yeast-yeast transitions in single flies.

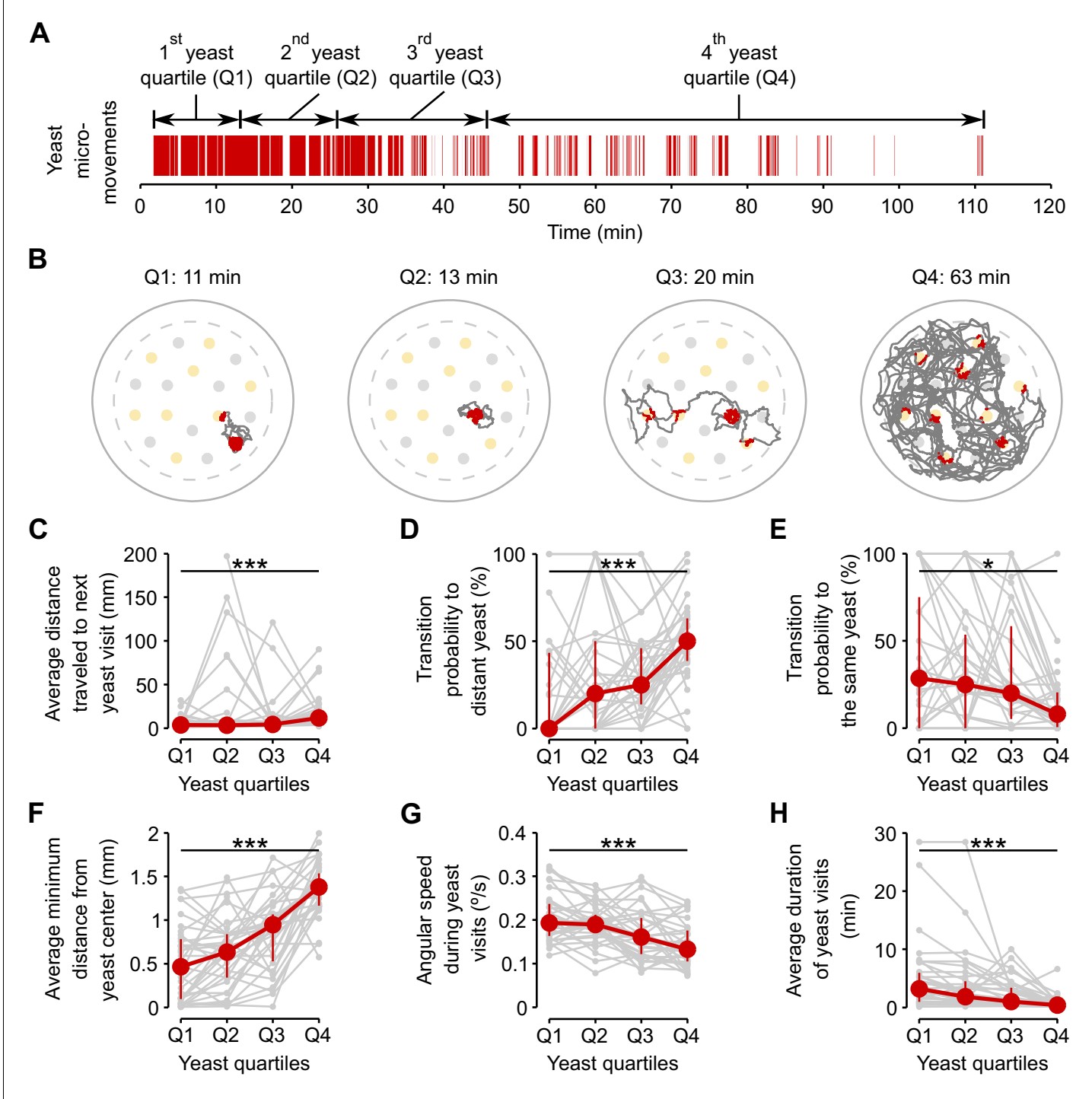

**Figure 6.** Flies dynamically adapt their exploitatory and exploratory behavior as their internal AA satiation changes. (A) Definition of yeast quartiles based on the total duration of yeast micromovements along the two hours of the video tracking assay for an example fly. Arrows indicate start and end points of each yeast quartile. Each yeast quartile consists of 25% of the time that the fly spent in yeast micromovements, but covers a different amount of absolute time in the assay for each fly, as shown in (B). (B) Example trajectories of head positions during each yeast quartile defined in (A). Red indicates the occurrence of a yeast micromovement. (C–H) Effect of yeast satiation on exploration (C–E) and exploitation (F–H) parameters, for mated AA-deprived flies, quantified during the four yeast quartiles of each fly. As the flies spend more time on yeast, the values of these parameters change towards the values of flies fed with a rich diet. *p<0.05, ***p<0.001, significance was tested by Wilcoxon rank-sum test.

The following figure supplement is available for figure 6:

*Figure 6 continued on next page*

*Figure 6 continued*

**Figure supplement 1.** Exploitation parameters in AA-deprived flies revert back to fully-fed values.

## ORs mediate efficient recognition of yeast as an appropriate food source

Starvation changes olfactory representations of food odors and these changes are thought to be required to find food efficiently (*Beshel and Zhong, 2013*; *Root et al., 2011*). As a proof of principle of how our setup could be used to uncover the neuronal mechanisms underlying foraging decisions, we decided to analyze the role of olfaction in nutrient homeostasis. Olfactory sensory neurons in *Drosophila* express two main types of chemosensory receptors: Odorant Receptors (OR) and Ionotropic glutamate receptors (IRs) (*Rytz et al., 2013*; *Vosshall and Stocker, 2007*). The OR type of olfactory receptors have been shown to significantly contribute to the olfactory detection of yeast over large distances (*Becher et al., 2010*; *Christiaens et al., 2014*) and are known to mediate physiological responses to yeast odors (*Libert et al., 2007*). We therefore focused on the function of these receptors in homeostatic yeast feeding by tracking the foraging behavior of flies lacking *Orco*, a co-receptor essential for OR function (*Larsson et al., 2004*). Unexpectedly, we observed that in general upon AA deprivation, *Orco* mutants showed a similar total duration of yeast visits as controls (*Figure 7A*). However, upon closer inspection of the time course of yeast visits, we observed that flies with impaired olfaction had a very long latency to engage in a long yeast visit when compared to the genetic controls (*Figure 7B–D*, see also *Figure 4A* and *Figure 7—figure supplement 1*). While *Orco* mutants needed around 25 min to enter into a high yeast exploitation 'mode' (median = 25.58 min, IQR = 15.05–30.06) the genetic controls required only 5–8 min to do so (*Figure 7C*).

Olfaction has been proposed to be important for the fly to locate food sources (*Root et al., 2011*). *Orco* mutants, however, have plenty of encounters with yeast during the latency period. This is clearly visible in the example trace (*Figure 7E*) where pink dots mark encounters with yeast patches. In fact, the number of encounters of *Orco* mutant flies with yeast patches was similar to, or even higher than, that of controls (*Figure 7F*). The increased latency also seems not to be due to an impairment in locomotion, as mutant flies walked as fast when outside the food patches as genetic controls (*Figure 7G*). These data indicate that in our assay, *Orco* mutant flies easily find yeast patches but fail to efficiently engage into long yeast visits.

If *Orco* mutant flies are inefficient in stopping at yeast patches, how do they manage to homeostatically compensate for the AA challenge? We observed that the duration of the first long visit (*Figure 7H*) and the probability of revisiting the same yeast patch (*Figure 7I*) were greater for the *Orco* mutants than for the controls. These results indicate that mutant flies were either more AA deprived than controls or compensated for their sensory deficit by displaying a generally higher exploitatory motivation. Taken together these results show that, in mated females, OR-mediated olfaction is necessary for efficient recognition of yeast as an appropriate resource but is not required to locate food patches at a short range or to achieve nutritional homeostasis.

## Octopamine mediates homeostatic postmating responses but not internal sensing of AA deprivation state

Neuromodulators are thought to be important in translating internal states into behavioral output (*Taghert and Nitabach, 2012*). While octopamine has been shown to mediate the postmating increase in yeast feeding (*Walker et al., 2015*), it has been proposed that it does not contribute to homeostatic changes in feeding behavior (*Yang et al., 2015*). We therefore decided to show that our setup could be used to test possible neuromodulatory effects of octopamine on yeast foraging, using mutants for the gene encoding Tyramine β-hydroxylase (TβH), an enzyme required for the biosynthesis of octopamine in the whole animal. As expected, we observed that in AA-deprived females, the drastic increase in the total duration of yeast visits after mating was greatly reduced in $T\beta h^{nM18}$ flies (*Figure 8A* and *Figure 8—figure supplement 1A*). Likewise, octopamine also seems to be required to elicit the full increase in the probability of stopping at yeast (*Figure 8B* and *Figure 8—figure supplement 1B*) and the full increase in the duration of yeast visits (*Figure 8C* and *Figure 8—figure supplement 1C*), reiterating our previous observation that these two parameters

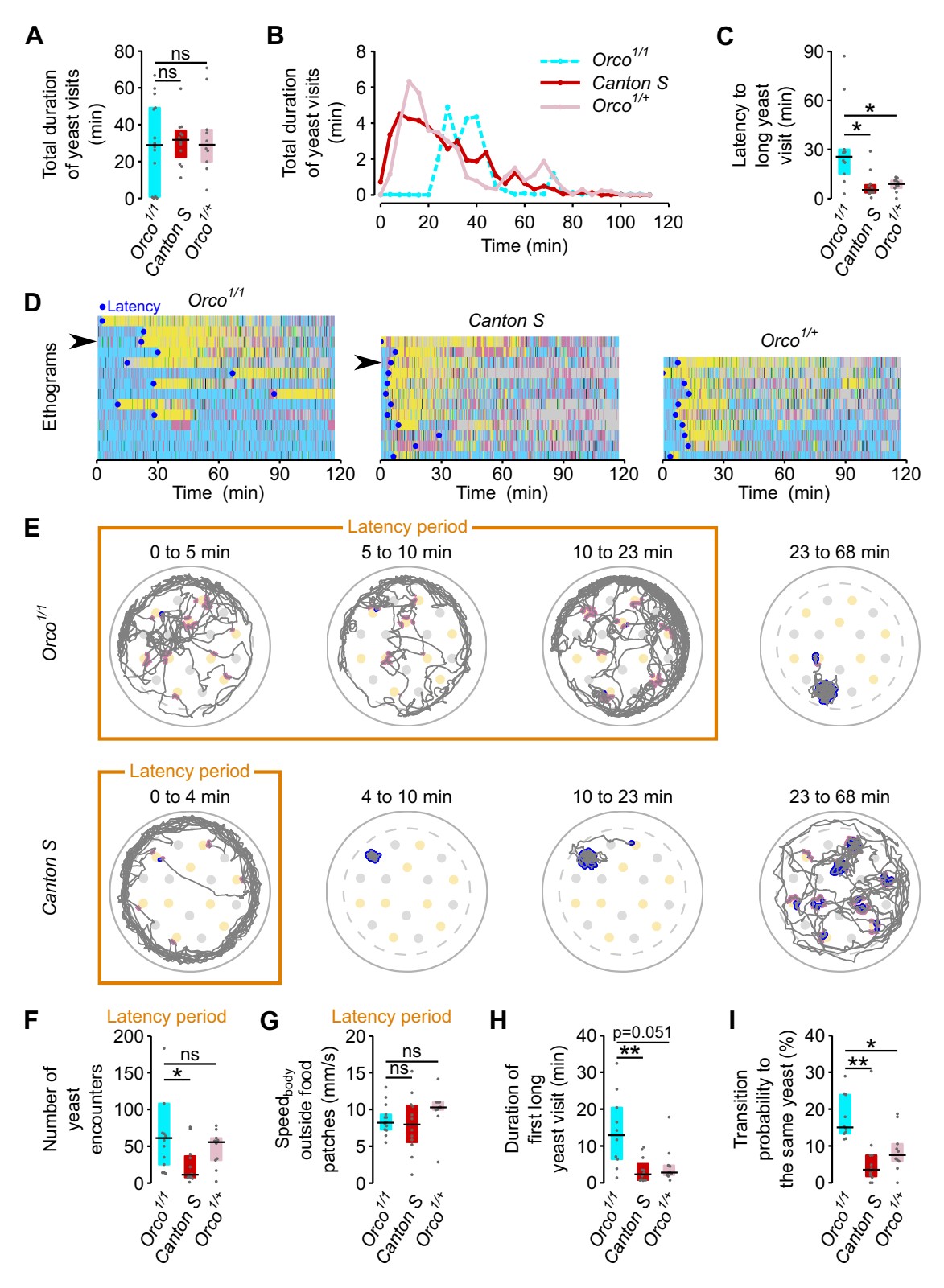

**Figure 7.** ORs mediate efficient recognition of yeast as an appropriate food source. (A) *Orco*[1/1] AA-deprived flies spend as much total time visiting yeast as AA-deprived control flies (n = 10–14). (B) Rolling median of the total duration of yeast visits using a 5 min window and a step of 4 min. (C) Effect of *Orco* mutation on the latency to engage in the first 'long' (≥30 s) yeast visit. (D) Behaviors displayed by *Orco*[1/1] and control flies, along the 120 min of the assay. Each row represents the ethogram of a single fly, following the same color code used in *Figure 1D*. Yellow: yeast micromovements.

*Figure 7 continued on next page*

*Figure 7 continued*

Black: sucrose micromovements. Pink: micromovements outside the food patches. Blue: walking bouts. Gray: resting bouts. Green: sharp turns. Blue circles indicate the latency (see **C**) of each fly. Arrows indicate example flies shown in (**E**). (**E**) Top: Example trajectory of head positions of an $Orco^{1/1}$ AA-deprived fly during the 23-min-long latency period (first three panels on the left) and during 45 min after the latency period (fourth panel). Bottom: Example trajectory of head positions of a *Canton S* AA-deprived fly during the 4-min-long latency period (first panel on the left) and from the latency point up to minute 68 (three panels on the right). Highlighted trajectory segments represent yeast encounters (pink) and yeast visits (blue). (**F–G**) Exploration and locomotor activity during latency period is not affected in $Orco^{1/1}$ flies as indicated by the number of yeast encounters (**F**) and the body centroid speed outside food patches (**G**). (**H**) The first long yeast visit is longer in $Orco^{1/1}$ flies than in control flies. (**I**) Probability of transition to same yeast patch is higher in $Orco^{1/1}$ flies than in control flies. *ns*, not significant ($p \geq 0.05$), *p<0.05, **p<0.01, significance was tested by Wilcoxon rank-sum test with Bonferroni correction.

The following figure supplement is available for figure 7:

**Figure supplement 1.** Yeast dynamics of *Orco* mutant flies.

are modulated by mating (*Figure 3*). To test whether octopamine was also required for mediating changes in yeast feeding behavior upon AA deprivation, we used the flyPAD technology. $T\beta h^{nM18}$ virgin flies were able to increase the number of sips after AA deprivation to a similar extent as control flies (*Figure 8D* and *Figure 8—figure supplement 1D*) showing that octopamine is not involved in translating the internal state of AA deprivation into increased yeast intake. Overall, these results confirm that the decisions to stop at a yeast patch and leave it are strongly modulated by mating. They also show that octopamine mediates these postmating responses towards yeast, but is not required to sense the internal AA deprivation state. These results provide a first step towards dissecting the role of octopamine in nutrient homeostasis.

## Discussion

In order to maintain nutrient homeostasis animals need to be able to adapt their nutrient preferences to their current state. But which behavioral alterations underlie such changes in preference? Here we use an automated video tracking setup to quantitatively capture the behavioral adaptations

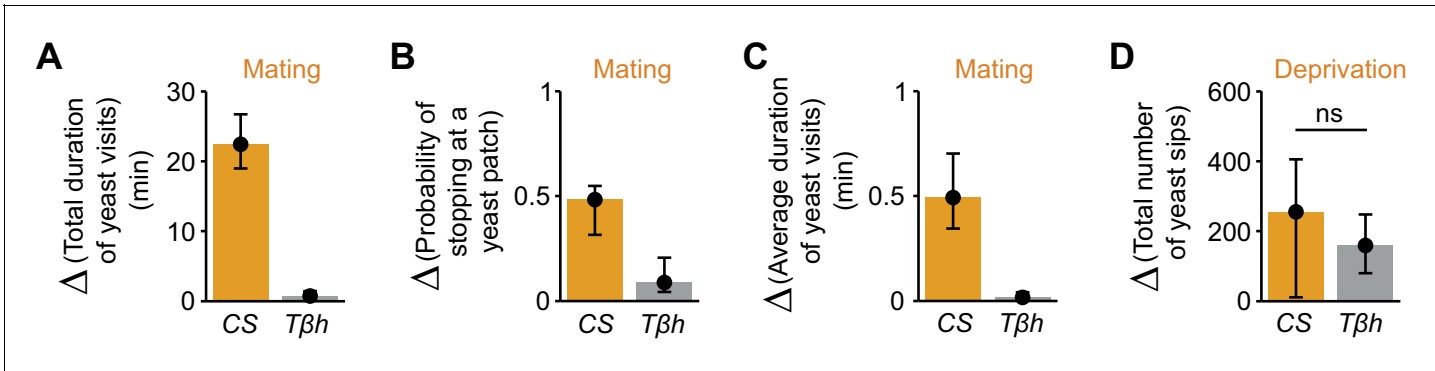

**Figure 8.** Octopamine mediates postmating response towards yeast but not internal sensing of AA deprivation state. (**A–C**) Effect of the $T\beta h^{nM18}$ mutation on the postmating change in foraging parameters, obtained from the video tracking assay after 1 hr: total duration of yeast visits (**A**), probability of stopping at a yeast patch (**B**) and average duration of yeast visits (**C**) for *Canton S* and $T\beta h^{nM18}$ females, both AA-deprived. Bars depict difference between median value of mated minus virgin groups for the correspondent parameter. Error bars show 5% and 95% bootstrap confidence intervals (n = 25–33). (**D**) Effect of $T\beta h$ mutation on the increase of yeast sips after AA deprivation in virgin females, quantified using the flyPAD setup. Bars depict difference between median values of AA+ (suboptimal) minus AA−deprived groups. Error bars show 5% and 95% bootstrap confidence intervals (n = 26–34). *ns*, not significant. (**A–C**) Show statistically significant differences between *Canton S* and $T\beta h^{nM18}$ females, as the confidence intervals don't overlap.

The following figure supplement is available for figure 8:

**Figure supplement 1.** Octopamine mediates postmating response to yeast.

to AA and mating state changes that allow the animal to maintain nutrient homeostasis. We started by separating the behaviors flies display towards food into discrete decisions: the decision to approach a food patch, the decision to stop at it, and the decision to leave it (*Figure 9*). Strikingly, mating and AA challenges induced compensatory behaviors towards yeast patches but not sucrose patches, indicating that the fly changes its exploitation decisions in a resource specific way. Furthermore, internal state modifications impact specific decisions to a different extent. While mating had a major effect on the probability of a fly leaving a yeast patch, AA challenges strongly increased the probability of stopping at a food patch. Nevertheless, the effect of AA deprivation on the decision to stop at a food patch was strongly dependent on mating suggesting that both internal states act synergistically to increase yeast intake. Furthermore, while full AA deprivation leads to a strong increase in yeast feeding when compared to flies kept on a suboptimal diet, the described decisions were not further altered by this drastic nutritional manipulation. There was, however, a considerable decrease in the latency to visit yeast patches for a long time and a general increase in parameters related to the 'eagerness' of the fly to exploit the resource (latency to engage on a yeast visit, locomotor activity on the patch and area of patch covered). Internal states, therefore, alter feeding in specific ways, allowing the fly either to spend more time on the food through the modulation of patch decisions, or to increase resource exploitation through the modulation of motivation without changing patch decisions. These specific changes allow the animal to dose its exploitatory behavior and hence the intake of nutrients over a large range (~17 fold) to match its current needs.

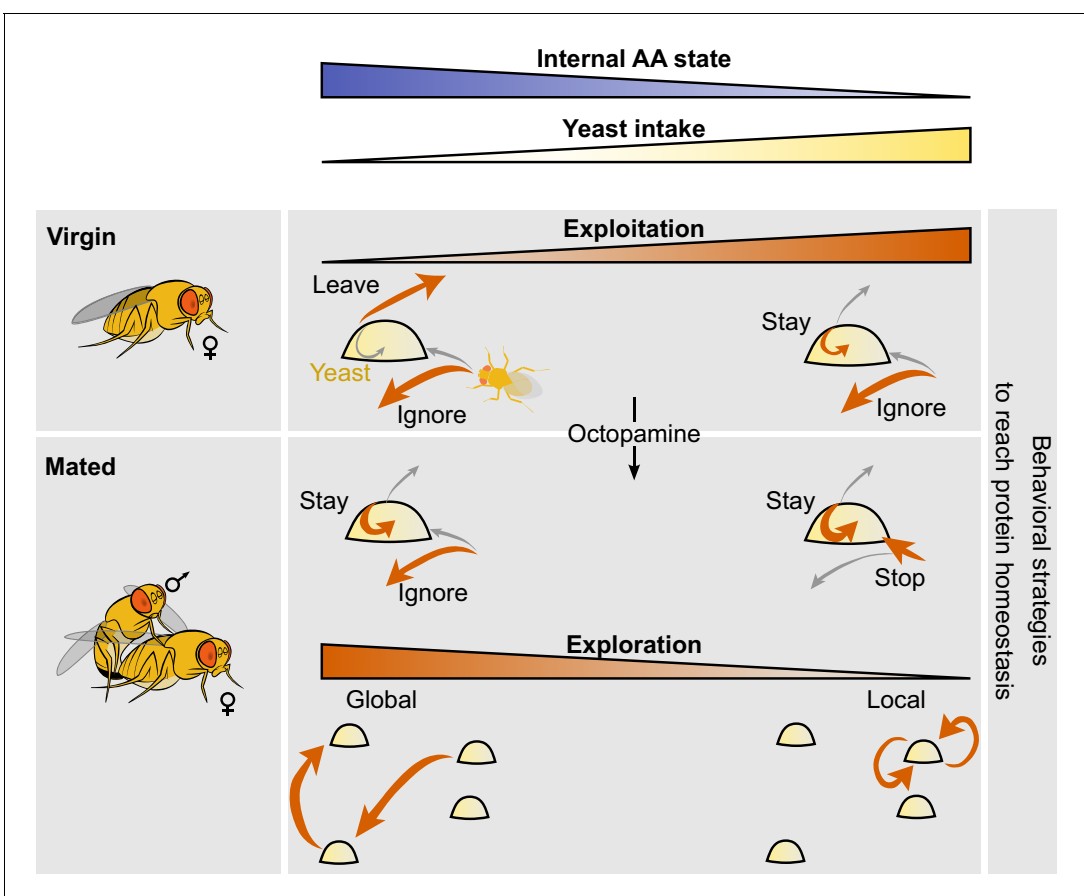

**Figure 9.** Model of behavioral strategies modulated by internal AA state. We propose a model in which virgin flies with high internal levels of AAs display low intake mostly ignore yeast patches upon encounter and have a high probability of leaving the yeast patch upon stopping at it. Internal AA levels decrease as a consequence of poor diets which induce a change in the leaving decision, inducing increased yeast intake. Octopamine mediates the postmating changes in the foraging decisions of stopping at the yeast patch and leaving it upon encounter. As the internal AA levels decrease in mated females, their exploration patterns switch from global exploration to local exploration and multiple returns to the same yeast patch.

The specific changes observed in the behavior upon alterations of internal state are in agreement with a modular organization of behavioral control. Such modularity has been previously described in the organization of motor output, such as locomotion (*Kiehn, 2016*), swimming (*Huang et al., 2013*), grooming (*Seeds et al., 2014*), and feeding (*Itskov et al., 2014*; *Walker et al., 2015*). This accumulated evidence suggests that the nervous system uses different mechanisms and hence circuits to change specific aspects of behavioral outcomes or decisions and that these changes add up to reach a specific goal. In agreement with this model, it has been shown that the impact of different starvation times on gustatory input relies on different mechanisms (*Inagaki et al., 2012*). Similarly, our data show that octopamine is specifically required for mediating the changes in yeast decisions observed upon mating but not upon AA deprivation (*Figure 9*). Nevertheless, some decisions, such as the decision to stop at a yeast patch, seem to be synergistically gated by both the mating and the AA state of the fly. It will be interesting to dissect how different internal states act at the circuit level to change behavioral decisions: do they act differentially on specific neuronal populations, or is the observed synergism a reflection of the different internal states acting on the same set of neurons?

While at the population level the effect of internal state manipulations led to stereotypical changes in behavior, the effect of internal state on the decisions implemented varied greatly at the individual level. This effect is reminiscent of the large individual differences observed in human physiology in response to identical diets (*Zeevi et al., 2015*). While such behavioral differences can stem from different metabolic states prior to the experiment, transgenerational effects in metabolism (*Öst et al., 2014*), or differences in the microbiota of the flies (*Broderick and Lemaitre, 2012*), there is a real possibility that they also reflect idiosyncrasies in behavior and metabolic susceptibilities to internal state changes at the individual level. Indeed, upon AA challenges, we observed that some flies increased their total time on yeast by having many short yeast visits, while some flies had fewer but longer visits. It will be interesting to investigate if these differences reflect behavioral idiosyncrasies, as observed before in many animals including *Drosophila* (*Buchanan and de Bivort, 2015*; *Dingemanse et al., 2010*; *Kain and de Bivort, 2012*). Differentiating between these two possibilities and identifying the physiological and circuit mechanisms leading to idiosyncrasies will be key to a better understanding of behavior. This is especially relevant for understanding metabolic conditions related to nutrition such as obesity.

In order to balance the intake of specific nutrients the animal should be able to specifically change its decisions towards the food source which contains the nutrient it needs. Our data show that this is indeed the case, pointing to a possible important contribution of chemosensory systems to nutrient decisions. Indeed, taste processing has been shown to be changed by the mating state of the animal and to contribute to the adaptation of behavioral decisions such as food choice (*Walker et al., 2015*) and egg laying site selection (*Hussain et al., 2016*). The contribution of olfaction to nutrient selection is less well understood. The sense of smell is thought to be mostly important for food search behavior (*Becher et al., 2010*), with starvation changing olfactory sensitivity to improve the finding of a food source (*Root et al., 2011*). Our data suggest that while olfactory-impaired mated flies are able to homeostatically increase yeast intake upon AA deprivation, OR-mediated olfaction still plays an important role in their capacity to do so. Interestingly, olfaction doesn't seem to be important for locating the food but for identifying yeast as an appropriate food source. These data suggest that flies use multimodal integration to decide which food to ingest. In humans, flavor, the integration of different sensory modalities such as taste and smell, is key to the perception of food (*Verhagen and Engelen, 2006*). Similarly, in mosquitoes olfaction acts together with other sensory cues to initiate a meal (*McMeniman et al., 2014*). Identifying the chemosensory basis for yeast feeding decisions might therefore be a powerful way to investigate the neuronal basis of flavor perception.

While one would expect that internal states increase food intake by changing exploitation decisions, their effects on exploratory behaviors in our paradigm are not trivial. Exploration is key for animals to find the resources they require and to acquire information about their surrounding environment (*Calhoun et al., 2014*; *Hassell and Shouthwood, 1978*; *Hills et al., 2015*). In our paradigm, however, animals do not need to search for resources as they are readily available. A key question then becomes why animals leave a food patch at all, especially when they are deprived of AAs (*Figure 9*). The fact that they still do so means that there is a value in leaving the current food patch, even if that one provides the urgently required nutrients to produce offspring and has not been depleted yet. We can only speculate that there must be an advantage in taking the 'risk' of

exploring unknown options and maybe identifying a better resource. Animals might often require other resources and leaving the current food patch might allow them to also explore the availability of these. Flies seem to nevertheless manage their exposure to uncertainty by tuning the spatial properties of their exploration. Their internal states not only define the probability of leaving a food patch, they also define if they will explore locally or more globally. The more deprived they are, the more local their exploratory pattern will be (*Figure 9*). Remarkably, while the leaving probability of flies pre-exposed to a suboptimal AA diet and a diet lacking AAs looks identical, their exploratory patterns are very different. For example, AA-deprived flies display a higher rate of returns to the same patch right after leaving it. Therefore, while the neuronal processes determining staying decisions seem not to be altered between both AA-challenged states, full AA deprivation must act on the circuits controlling exploration to strongly increase the probability of revisiting the patch the fly just left. This allows the fly to de facto stay longer on the same food patch without changing its leaving decisions. We would like to propose that the apparent separate regulation of these two aspects of the fly behavior suggests that there are two separate internal state sensing processes regulating exploitation and exploration decisions. The combination of both behavioral and circuit modules would allow the fly to trade off the requirement to exploit specific resources and the 'risk' of exposing itself to resources of unknown or lower quality. Furthermore, it is interesting to note the similarity between the revisits to the same food patch we observed upon strong AA deprivation and the 'dances' observed by Vincent Dethier in the blowfly (*Dethier, 1976*). Both phenomena are examples of how animals regulate their search behavior and exposure to uncertainty by modulating the local dynamics of their exploratory behavior, in a state-dependent manner. While the budget theory is a classic aspect of foraging theory, it has recently started to be reassessed. It is mainly controversial if energy-deprived animals, including humans, are more or less risk-prone (*Kacelnik and El Mouden, 2013*). Our data suggest that the exploratory behavior of AA-deprived animals minimizes their exposure to uncertainty. The description of how different aspects of risk management are implemented at the behavioral level opens up the opportunity to identify the circuit mechanisms by which internal states control exploration-exploitation trade-offs and therefore how animals decide how much to expose themselves to the unknown.

The success of neurogenetics has relied to a large extent on the use of simple binary end-point behavioral assays to perform large-scale unbiased screens (*Leitão-gonçalves and Ribeiro, 2014*; *Ugur et al., 2016*; *Vosshall, 2007*). This approach has allowed the field to make important contributions to the molecular and circuit basis of innumerable phenomena, including learning and memory (*Heisenberg, 2015*), chronobiology (*Konopka and Benzer, 1971*), innate behaviors (*Demir and Dickson, 2005*; *Yapici et al., 2008*), and sensory physiology (*Larsson et al., 2004*). While identifying these cornerstones of neuroscience was crucial, we are now in a position to start understanding how these mechanisms act at the circuit level to perform more complex computations such as the ones used during decision-making and exploration. This endeavor requires the use of a richer and dynamic description and analysis of behavior (*Gomez-Marin et al., 2014*). We used a combination of computer vision (*Anderson and Perona, 2014*) and a quantitative, automated capacitance-based behavioral assay (*Itskov et al., 2014*) with internal state and genetic manipulations to characterize and identify the behavioral changes allowing the animal to achieve homeostasis. It is interesting to consider that while we identify an important role of OR-mediated olfaction in nutrient decision-making, this would not have been possible using end-point analyses, as the animal manages to compensate for its sensory challenge using alternative means. The use of dynamic, quantitative descriptions of complex behavior therefore enables neuroscientists to decompose these into discrete processes, opening up the possibility to go beyond assigning circuits and molecules to general behaviors to start explaining how they act to control the generation of complex cognitive processes.

## Materials and methods

### *Drosophila* stocks, genetics and rearing conditions

Unless stated otherwise all experiments were performed with *Canton S* female flies. *Canton S* flies were obtained from the Bloomington stock center. *Orco*$^{1/1}$ flies were a kind gift of Sofia Lavista-Llanos from the Hansson laboratory (*Larsson et al., 2004*). *Orco*$^{1/+}$ flies were obtained by crossing *Canton S* virgins with *Orco*$^{1/1}$ males. *Tβh*$^{nM18}$ flies were a kind gift of Scott Waddell (*Monastirioti et al.,*

*1996*). Fly rearing, maintenance, and behavioral testing were performed at 25°C in climate-controlled chambers at 70% relative humidity in a 12 hr-light-dark cycle. All experimental and control groups were matched for age and husbandry conditions.

## Media compositions

The standard yeast-based medium (YBM) contained, per liter, 80 g cane molasses, 22 g sugar beet syrup, 8 g agar, 80 g corn flour, 10 g soya flour, 18 g yeast extract, 8 ml propionic acid, and 12 ml nipagin (15% in ethanol) supplemented with instant yeast granules on the surface. To ensure a homogenous density of offspring among experiments, fly cultures were always set with five females and three males per vial and left to lay eggs for three days. Flies were reared in YBM until adulthood. Three different types of holidic medium were used as described previously (*Piper et al., 2013*) using the following formulations: 50S200NYaa (*AA+ rich*), 50S200NHUNTaa (*AA+ suboptimal*) and 50S0N (*AA−*). Composition of the media is as described in *Piper et al., (2013)*, without food preservatives and only differ in amino acids composition. The proportion of amino acids in 50S200NYaa diet is adjusted to match that in yeast and was considered a rich diet maximizing egg laying (*Piper et al., 2013*). The detailed holidic media compositions can be found in *Table 1*.

## Behavioral assays

Groups of 9 to 11 newly hatched (0–6 hr old) female flies of the indicated genotypes were transferred to vials containing fresh standard yeast-based medium (YBM). Three days later, all vials were transferred to fresh standard medium and 4 males were added to half of the vials to obtain mated female flies. After two more days, all vials were transferred once again to fresh standard YBM. On the sixth day, all vials were transferred to fresh holidic media. Flies were left to feed *ad-libitum* for three days on the holidic media and then tested in the video tracking or flyPAD setups. Single flies were tested in individual arenas that contained two kinds of food patches: yeast (180 g/L) and sucrose (180 g/L), each mixed with 0.75% (tracking) or 1% (flyPAD) agarose. Flies were individually transferred to the arenas by mouth aspiration and allowed to feed for 1 (flyPAD) or 2 (tracking) hours, except for the tracking experiment with $T\beta h^{nM18}$ mutant flies, which lasted 1 hr. flyPAD data were acquired using the Bonsai framework (*Lopes et al., 2015*) and analyzed in MATLAB (Mathworks, Natick, MA) using custom-written software, as described (*Itskov et al., 2014*). To avoid patch exhaustion before the end of the tracking assays, each circular patch contained 5 µL of food with a diameter of approximately 3 mm. After each assay, the tracking arenas were washed with soap, rinsed with 70% ethanol, and finally with distilled water.

Videos that had more than 10% of lost frames (due to technical problems during acquisition) were excluded from the analysis. No further data was excluded. The experiment that compares the conditions *AA+ suboptimal* and *AA−* (results shown in *Figures 1–6*) was performed 3 times independently, which means that an independent set of individuals (n=15–35, shown in the corresponding figure legend) was reared and tested under the corresponding conditions. The experiment comparing *AA+ rich* vs *AA−* was performed two times independently. The experiments comparing $T\beta h^{nM18}$ or *Orco* mutant flies with their corresponding controls were performed once with the sample size indicated in the corresponding figure legend. We confirmed that the claims made in this manuscript held for every experimental replicate. We never tested the same individual more than once.

## Behavioral box and arena design

The behavioral arenas for the video tracking (*Figure 1B*) were designed and manufactured in-house using a laser-cutter and a milling machine. Material used for the base was acrylic and glass for the lid. The outer diameter of the arena was 73 mm. The inner area containing food patches was flat and had a diameter of 50 mm and a distance to the lid of 2.1 mm. To allow a top-view of the fly throughout the whole experiment, the outer area of the arena had 10° of inclination (*Simon and Dickinson, 2010*) and the glass lid was coated with 10 µL of SigmaCote the night before the assays. Food patches were distributed in two concentric circles equidistantly from the edge. Furthermore, sucrose and yeast patches were alternated such that from a given food patch, there was at least one adjacent yeast and one adjacent sucrose patch. The radius of each food patch was approximately 1.5 mm. The minimum distance between the centers of two adjacent food patches is 10 mm. White LEDs 12V DC (4.8 watt/meter), were used for illumination of the arenas. They were placed under the

**Table 1.** Composition of holidic medium.

| | Ingredient | Stock | Amount per liter |
|---|---|---|---|
| Gelling agent | Agar | | 20 g |
| Sugar | Sucrose | | 17.12 g |
| Amino acids for 50S200NHUNTaa* | L-isoleucine | | 1.82 g |
| | L-leucine | | 1.21 g |
| | L-tyrosine | | 0.42 g |
| Amino acids for 50S200NYaa* | L-isoleucine | | 1.16 g |
| | L-leucine | | 1.64 g |
| | L-tyrosine | | 0.84 g |
| Metal ions | $CaCl_2.6H_2O$ | 1000x: 250 g/l | 1 ml |
| | $CuSO_4.5H_2O$ | 1000x: 2.5 g/l | 1 ml |
| | $FeSO_4.7H_2O$ | 1000x: 25 g/l | 1 ml |
| | $MgSO_4$ (anhydrous) | 1000x: 250 g/l | 1 ml |
| | $MnCl_2.4H_2O$ | 1000x:1 g/l | 1 ml |
| | $ZnSO_4.7H_2O$ | 1000x: 25 g/l | 1 ml |
| Cholesterol | Cholesterol | 20 mg/ml in Ethanol | 15 ml |
| Water | Water (milliQ) | 1 l minus combined volume to be added after autoclaving | |
| **Autoclave 15 min at 120°C. All additions below should be performed using sterile technique** | | | |
| Amino acids for 50S200NHUNTaa* | Essential amino acid stock solution | 8 g/l L- arginine monohydrochloride<br>10 g/l L-histidine<br>19 g/l L- lysine monohydrochloride<br>8 g/l L-methionine<br>13 g/l L-phenylalanine<br>20 g/l L-threonine<br>5 g/l L-tryptophan<br>28 g/l L-valine | 60.51 ml |
| | Non-essential amino acid stock solution | 35 g/l L-alanine<br>17 g/l L-asparagine<br>17 g/l L-aspartic acid sodium salt monohydrate<br>0.5 g/l L-cysteine hydrochloride<br>25 g/l L-glutamine<br>32 g/l glycine<br>15 g/l L-proline<br>19 g/l L-serine | 60.51 ml |
| | Sodium glutamate stock solution | 100 g/l L-glutamic acid monosodium salt hydrate | 15.13 ml |
| Amino acids for 50S200NYaa* | Essential amino acid stock solution | 23.51 g/l L-arginine monohydrochloride<br>11.21 g/l L-histidine<br>28.70 g/l L-lysine monohydrochloride<br>5.62 g/l L-methionine<br>15.14 g/l L-phenylalanine<br>21.39 g/l L-threonine<br>7.27 g/l L-tryptophan<br>22.12 g/l L-valine | 60.51 ml |
| | Non-essential amino acid stock solution | 26.25 g/l L-alanine<br>13.89 g/l L-asparagine<br>13.89 g/l L-aspartic acid sodium salt monohydrate<br>30.09 g/l L-glutamine<br>17.89 g/l glycine<br>9.32 g/l L-proline<br>12.56 g/l L-serine | 60.51 ml |
| | Sodium glutamate stock solution | 100 g/l L-glutamic acid monosodium salt hydrate | 18.21 ml |
| | Cysteine stock solution | 50 g/l L-cysteine hydrochloride | 5.28 ml |

*Table 1 continued on next page*

*Table 1 continued*

| | Ingredient | Stock | Amount per liter |
|---|---|---|---|
| Vitamins | Vitamin solution | 125x:<br>0.1 g/l thiamine hydrochloride<br>0.05 g/l riboflavin<br>0.6 g/l nicotinic acid<br>0.775 g/l Ca pantothenate<br>0.125 g/l pyridoxine hydrochloride<br>0.01 g/l biotin | 14 ml |
| | Sodium folate | 1000x: 0.5 g/l | 1 ml |
| Base | Buffer | 10x:<br>30 ml/l glacial acetic acid<br>30 g/l $KH_2PO_4$<br>10 g/l $NaHCO_3$ | 100 ml |
| Other nutrients | | 125x:<br>6.25 g/l choline chloride<br>0.63 g/l myo-inositol<br>8.13 g/l inosine<br>7.5 g/l uridine | 8 ml |

* To prepare the 50S200NHUNTaa diet, use the values shaded in blue. To prepare the 50S200NYaa diet, use the values shaded in orange.

arenas, as backlight illumination and on the walls of the behavioral box, surrounding the arenas, as shown in *Figure 1A*. A white cardboard arch was used to improve illumination to reflect light towards the arenas (*Figure 1A*). Three fly arenas were recorded simultaneously from the top using a video camera (Genie HM1400 camera, Teledyne DALSA, Canada; frame acquisition rate: 50 fps) connected to a desktop computer using a Gigabit Ethernet connection.

## Tracking algorithm

Body centroid positions and major axis of the fly body in each frame were extracted using custom off-line tracking algorithms written in Bonsai (*Lopes et al., 2015*) and Matlab (Mathworks). The arena diameter in the video was measured to find the correspondence between pixels in the video and mm in the real world (1 pixel = 0.155 mm). The typical length of the major axis of the fly body in a video was 19 pixels (~3 mm). Video acquisition was made with slight overexposure to obtain a strong contrast between the fly and the arena. Since the fly body was the darkest object in the arena, a pixel intensity threshold was used to obtain the centroid and orientation of the fly blob. The head position was extracted using custom MATLAB (Mathworks) software. Head position in the first frame was manually selected. From there on, the head position is automatically propagated to the consecutive frames using a proximity rule (*Gomez-Marin et al., 2011*). This rule, however, does not hold during a jump of the fly. Therefore, in addition to the proximity rule, for the intervals in-between jumps, the head position was automatically corrected using the fact that flies walk forward most of the time. Manual annotation of 510 inter-jump-intervals revealed that 98% were correctly classified. All the body and head centroid tracking data generated in this study are available for download from the Dryad repository (*Corrales-Carvajal et al., 2016*).

## Behavioral classification

Raw trajectories of head and body centroids were smoothed using a Gaussian filter of 16 frames (0.32 s) width. The width was chosen empirically by comparing the raw and smoothed tracks. The speed was measured from the smoothed coordinates by calculating the distance covered from the current frame and the next frame, divided by the time between them (0.02 s). Similarly, the angular speed was measured by calculating the difference between the heading angle from the current frame and the next frame, divided by the time between them. The heading angle for this calculation was obtained from the head and tail smoothed centroids. Walking and non-walking instances were classified applying a 2 mm/s threshold in the head speed, based on the distribution of head speed for AA-deprived flies in *Figure 1—figure supplement 1A* and previous studies (*Martin, 2004*; *Robie et al., 2010*). The head speed used was also smoothed using a Gaussian filter of 60 frames

(1.2 s) to avoid rapid changes in classification around the thresholds. Sharp turns were classified when a local maximum in the angular speed exceeded a 125°/s threshold, as long as the body centroid speed was below 4 mm/s. A wider Gaussian filter (width of 2.4 s) was applied to the head speed to classify resting bouts, using a threshold of 0.2 mm/s. The remaining events during the non-walking segments that were not classified as resting were classified as micromovements.

## Food encounters, micromovements, and visits

Manual annotation of 107 feeding events showed that when the head position was at 3 mm or less from the center of the food patch, flies were already close enough to have leg contact. Initially, *encounters* with a food patch were defined as the moments in which the fly crossed this 3 mm distance threshold. To avoid misclassifying the transient head movement associated with grooming or feeding around this threshold as new encounters, consecutive encounters were merged when the total displacement of the head in any direction was lower or equal than 2 pixels (0.31 mm) during the time elapsed in-between the encounters. From each feeding event, the distance from the head of the fly to the center of the patch was also captured. Since 95% of the first proboscis extensions happened below 2.5 mm, this was the selected distance threshold to define *yeast* and *sucrose micromovements* (*Figure 1—figure supplement 1D*). In this way, food micromovements were defined as the time in which flies were classified in a micromovement (see definition in previous section) and their head was simultaneously inside a circle of 2.5 mm around the food patch (see gray dashed line in *Figure 1D* inset). The two pixels displacement rule used in the definition of encounters was also applied here to avoid definitions of false new micromovements. A *visit* was defined as a series of consecutive food micromovements (already corrected for small displacements) in which the head distance to the center of the food patch was never larger than 5 mm during the time elapsed in-between the food micromovements (*Figure 1D* inset). 5 mm is the maximum radius of non-overlapping circles around the food patches (see gray dashed line in main trajectory of *Figure 1D*). This 5 mm threshold was also used to merge consecutive encounters (consecutive encounters were merged if the head distance to the center of the food patch was never larger than 5 mm during the time in-between encounters). In this way, for every visit there is an encounter, but there can be an encounter and no visit if the fly doesn't stop at the food patch (food micromovement).

## Exploitation, exploration and locomotor activity parameters

All of these parameters, unless specified otherwise, were calculated for each fly and for the whole duration of the assay.

1. **Yeast** (or sucrose) **micromovements**: Events in which the fly was micromoving (0.2 mm/s < head speed < 2 mm/s, see *Behavioral classification* section for details) on the food patch (head position $\leq$ 2.5 mm from the center of the food patch).
2. **Total duration of yeast** (or sucrose) **micromovements** (**min**): Sum of the durations of all yeast (or sucrose) micromovements. Initially calculated in frames and converted to minutes by dividing by the frame rate (50 frames per second) and dividing by 60.
3. **Fraction of yeast non-eaters**: Number of flies with a total duration of yeast visits lower than 1 min divided by the total number of flies in that internal state condition.
4. **Coefficient of variation:** Standard deviation divided by the mean of the total duration of yeast micromovements for each internal state condition.
5. **Cumulative time of yeast micromovements** (**min**): Cumulative sum of frames in which the fly was in a yeast micromovement, converted to minutes as described for parameter 2.
6. **Yeast** (or sucrose) **visits:** Series of consecutive food micromovements in which the head distance to the center of the food patch was never larger than 5 mm during the time elapsed in-between the food micromovements.
7. **Total duration of yeast visits** (**min**): Sum of the durations of all yeast visits. Durations of visits were calculated similarly to parameter 2.
8. **Number of yeast encounters**: Sum of all yeast encounters.
9. **Rate of yeast encounters**: Sum of all yeast encounters divided by the time spent walking outside the food patches.
10. **Probability of stopping at a yeast patch**: Number of encounters with yeast that contained at least one yeast micromovement, divided by the total number of yeast encounters.
11. **Average duration of yeast visits** (**min**): Sum of all the durations of yeast visits divided by the total number of yeast visits.

12. **Number of yeast visits**: Sum of all yeast visits.
13. **Rolling median of total duration of yeast visits** (**min**): Sum of the duration of all yeast visits that occurred within a 5 min sliding window with a step of 4 min.
14. **Latency to long yeast visit** (**min**): Time elapsed from the beginning of the assay until the fly engages in a yeast visit which is at least 30 s long.
15. **Average minimum distance from yeast** (**mm**): Average of the minimum distance from the head to the center of the yeast patch for each visit, across all yeast visits.
16. **Average area covered during yeast visits** (**pixels**): Average across all yeast visits of the number of different pixels covered by the head of the fly during each yeast visit.
17. **Speed$_{body}$ during yeast visits** (**mm/s**): Average of body centroid speed across all frames in which the fly was inside a yeast visit. Calculation of speed was as described in the *behavioral classification* section and then smoothed using a Gaussian filter of 60 frames (1.2 s).
18. **Angular speed during yeast visits** (**°/s**): Average of angular speed across all frames in which the fly was inside a yeast visit. Calculation of angular speed was as described in the *behavioral classification* section.
19. **Transition probability to a distant yeast patch**: Number of visits to a distant yeast patch divided by the total number of yeast visits. Visits to distant yeast patches were of two kinds: either the distance between the previous and current patch centers was >16 mm or the distance from the fly head to the center of the previous patch was >16 mm at any point during the inter-visit-interval. Only transitions between *visited* yeast patches were considered.
20. **Transition probability to an adjacent yeast patch**: Number of visits to an adjacent yeast patch divided by the total number of yeast visits. Two patches were defined as adjacent if the distance between their centers was ≤16 mm. Only transitions in which the distance from the fly head to the center of the previous patch was ≤16 mm during the whole inter-visit-interval were considered. Otherwise, the transition was classified as to a distant yeast patch (see parameter 19). Only transitions between *visited* yeast patches were considered.
21. **Transition probability to the same yeast patch**: Number of visits to the same yeast patch divided by the total number of yeast visits. Visits to the same yeast patch were those in which the previous visit happened in the same patch as the current visit. Only transitions in which the distance from the fly head to the center of the previous patch was ≤16 mm during the whole inter-visit-interval were considered. Otherwise, the transition was classified as to a distant yeast patch (see parameter 19). Only transitions between *visited* yeast patches were considered. The values of the transition probabilities depicted as pie charts in *Figure 5A–C* are the medians shown in panels D-F scaled so they sum to 100%.
22. **Distance traveled to next yeast visit** (**mm**): Average distance covered from the end of a visit to any food patch (yeast or sucrose) to the beginning of the next visit to a yeast patch.
23. **Yeast quartiles**: The sum of the durations of all the yeast micromovements from the latency point (see parameter 14) onwards was considered as 100% of yeast time for a given fly. First yeast quartile (Q1) was the time elapsed between the latency point and the 25% of yeast time for that fly. Q2 was the time elapsed between 25% and 50% of yeast time for that fly. In the same way, Q3 went from 50% to 75% and Q4 from 75% to 100% of yeast time for that fly.
24. **Speed outside food patches** (**mm/s**): Similar to parameter 17, but for all frames in which the fly was not engaged in a food visit. In *Figure 7G*, this parameter was calculated only for the latency period (parameter 14).

## Acknowledgements

We thank Sofia Lavista-Llanos (Hansson Lab), Scott Waddell and Carolina Rezával (Goodwin Lab) for sharing fly stocks. Stocks obtained from the Bloomington Drosophila Stock Center (NIH P40OD018537) were used in this study. We thank Célia Baltazar for technical assistance in running some of the tracking and flyPAD experiments; Pavel Itskov for assistance with analysis code for fly-PAD data; Ana Machado for contributions to early phases of this project; Alex Gomez-Marin, Gonçalo Lopes, José Cruz and Ricardo Ribeiro for help in improving the tracking algorithms; and FABLAB-EDP (http://fablabedp.edp.pt) for kindly providing access to the milling machine used for the arenas manufacturing. We also thank Christian K Machens, Gonzalo G de Polavieja, Samuel SJ Walker and members of the Behavior and Metabolism laboratory for helpful discussions and comments on the manuscript, as well as Gil Costa for his help with illustrations.

# Additional information

## Funding

| Funder | Grant reference number | Author |
|---|---|---|
| Fundação para a Ciência e a Tecnologia | PTDC/BIA-BCM/118684/2010 | Carlos Ribeiro |
| Human Frontier Science Program | RGP0022/2012 | Aldo A Faisal<br>Carlos Ribeiro |
| Champalimaud Foundation | | Verónica María Corrales-Carvajal<br>Carlos Ribeiro |
| Fundação para a Ciência e a Tecnologia | Graduate Student Fellowship, SFRH/BD/51113/2010 | Verónica María Corrales-Carvajal |

The funders had no role in study design, data collection and interpretation, or the decision to submit the work for publication.

## Author contributions

VMC-C, Technical development of the project, Conception and design, Acquisition of data, Analysis and interpretation of data, Drafting or revising the article; AAF, Initial conceptual and technical development of the project, Conception and design; CR, Initial technical development of the project, Supervision of the project, Conception and design, Analysis and interpretation of data, Drafting or revising the article

## Author ORCIDs

Verónica María Corrales-Carvajal, http://orcid.org/0000-0002-3813-5790
Carlos Ribeiro, http://orcid.org/0000-0002-9542-7335

# Additional files

## Major datasets

The following dataset was generated:

| Author(s) | Year | Dataset title | Dataset URL | Database, license, and accessibility information |
|---|---|---|---|---|
| Corrales-Carvajal VM, Faisal AA, Ribeiro C | 2016 | Data from: Internal states drive nutrient homeostasis by modulating exploration-exploitation trade-off | http://dx.doi.org/10.5061/dryad.s58qh | Available at Dryad Digital Repository under a CC0 Public Domain Dedication |

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
