## [Decision Letter]

Thank you for submitting your article "Internal states drive nutrient homeostasis by modulating exploration-exploitation trade-off" for consideration by *eLife*. Your article has been favorably evaluated by K VijayRaghavan as the Senior Editor and three reviewers, including Steve Simpson (Reviewer #2) and a member of our Board of Reviewing Editors.

The reviewers have discussed the reviews with one another and the Reviewing Editor has drafted this decision to help you prepare a revised submission..

Summary:

All of the reviewers applaud the rigor of the quantitative approach employed in this work, and find value in the comprehensive analysis of *Drosophila* behavior in the face of specific nutritional challenges. The results were seen as both interesting and useful to a broad range of readers.

Essential revisions:

It is felt that the link to neural mechanism is rather a 'proof of principle', and does not go deep into the mechanism as may be possible with this model system. It was not felt that this is a critical flaw, but rather that the limitations need to be more clearly discussed, providing an opportunity to set the scene as to where research of this type could (and likely will) be directed in future.

*Reviewer #1:*

This paper addresses the very important challenge of identifying key drivers of nutritional decision-making using *Drosophila* as a tractable model organism. Furthermore, it employs advanced tracking to quantify the motion and precise feeding decisions of individuals among multiple resource patches.

This analysis shows that flies sense deficits in amino acids and can adjust their motion and feeding behavior to compensate, and that this is impacted by their mating state. Given that the experimental design was such as to remove (to a great degree) the effects of search, and rather to focus on the behavior of flies once on a patch, the analysis focused on the decisions to engage and to leave (as opposed to find) patches. The analysis includes highly detailed information as to how flies move on the patches.

An area which needs improving, in my mind, however, is at it relates to the neural mechanisms at work. This has the potential to be the most interesting part of the paper (and the one I was looking forward to getting to!), but in the end it provides useful, but indirect, evidence of the actual mechanisms at work. For example, the experiments suggest the importance of octopamine, but they do not get into the regions of the brain (or even if it is in the brain) important, and I think that while this section is helpful it is more of a start than a conclusion to the story, and perhaps the authors could make more clear what is, and what is not, shown by these data.

In summary I find this a highly rigorous and useful paper.

*Reviewer #2:*

The authors have provided an extremely detailed behavioural analysis of nutrient-specific foraging behaviour in *Drosophila*, coupled with two experiments looking at the involvement of olfaction and octopamine in different stages of the control of protein homeostasis. Although the manuscript is extensive, the narrative is interesting and the illustrations beautifully composed. The paper adds significantly to the literature on appetite control and nutrient regulation. I have a few somewhat substantive issues.

Abstract: change "deeply" to "profoundly" or similar. I would also remove the second sentence, which is over-egging things and not strictly true.

I felt the term "engaging the food" to be a little awkward (begs the question "Engaging food in what?"). "Stop at" is what the data indicate – or "arrested at the food". I pick up on the precision of language a little more below.

In the subsection “The lack of dietary AAs increases exploitation and local exploration of yeast patches”. This is more by way of a comment than a suggestion: the point is introduced and to some extent tested, but it is worth pointing out somewhere that the volume ingested during a feeding bout (meal) is a product of the duration of the feeding bout and the rate at which food is ingested across the meal. Meal size (and its volumetric and other determinants) has been shown in some insect systems to be positively correlated with ingestion rate and affected by common inputs (both homeostatic and non-homeostatic) (e.g. Animal Behav. 36: 1216-27). Hence, meal duration (which is estimated in the present set up) tends to be a more conserved variable than meal size (a fly can eat more by ingesting faster over the same meal duration). Changes in ingestion rate can result from greater amplitude of feeding central pattern generator (and hence motor) output, or greater frequency of rhythm, or through a change in intra-meal burst and interval structure. The data reported address and show a contribution from the latter, but changes in both amplitude and frequency of CPG output are also known.

In the fourth paragraph of the subsection “The lack of dietary AAs increases exploitation and local exploration of yeast patches”.– 'intake' rather than 'use', which is a vague term. There are numerous other instances throughout the manuscript of faintly teleological terminology – "decision", "engage", "exploitation", "exploration", "choose to visit distant patches", etc. In such a beautiful analysis of behaviour as this, I would rather see clear, objective description. Maybe I'm being a pedant on this, but J.S. Kennedy (The New Anthropomorphism, 1993, CUP) was spot on in my view.

The one counter-intuitive finding is that reported in the subsection “Amino acid challenges reduce global exploration and increase revisits to the same yeast patch”. My understanding is that theory and experiment (Levy flight; intrinsic (local) search vs. extrinsic search (ranging); Dethier's fly dance, etc.) predicts that flies should stay nearby "high quality" patches and roam more extensively when on "lower quality" patches. Staying nearby to an imbalanced food source when in a state of AA imbalance and roaming globally when AA balanced appears contrary to expectations and suggests something interesting, perhaps about the evolved predicted spatial distribution of protein resources in the environment, e.g. complementary imbalanced AA sources are likely to be close by to one another. In the last paragraph of the aforementioned subsection there is an unconvincing statement on this issue, which needs some more thought I think.

Subsection “ORs mediate efficient recognition of yeast as an appropriate food source”. This section could be set up by referencing back to the observation reported earlier that arrival rates at food sources was not impacted by state – implying that olfactory cues acting over a distance are not involved, but close-range chemosensory cues (short-range olfactory and gustatory) are more likely. Would help build the narrative.

The Discussion is where you need to address the matter mentioned above in the subsection “Amino acid challenges reduce global exploration and increase revisits to the same yeast patch”. The blowfly dance results seem at odds with present findings – flies dance more vigorously and stay longer locally after stimulation with a higher concentration food source. The same is the case for bees and other animals (William Bell's work – e.g. see his 1990 book). It is true that greater levels of deprivation elicit greater local dancing to a given food; but still the effect is strongly dependent on food quality relative to state.

*Reviewer #3:*

Ribeiro's group uses this paper to establish a new assay to assess the relationships between internal state defined by the interaction between sexual experience (mated or virgin) and metabolic experience (more or less amino acids in the medium) on behaviour (explore or exploit). I like the analysis very much and I think it will be a welcome addition to the literature. Their method includes potential to dissect decision-making at the level of behavior and offers a world of new possibilities for evaluating the contribution of genes, development, and physiology to behavior. They note and promise that they are providing insight into mechanism as well and this part of the paper pales when compared to the behavioral analysis. The nod to mechanism occurs via the manipulation of *Orco* to demonstrate a role for smell in the behavioural strategy used by mated females. Not clear that it plays a role for the virgins because there are no data and it seems less likely but entirely possible that olfaction is used to detect or evaluate sugars. I do not think more experiments need to be performed but I think the *Orco* data provide more of a proof of principle than an exposition of mechanism and the paper would be improved if these data were presented that way. This same point applies to the tyramine β hydroxylase experiments. The paper is written as if these data provide insight into the brain, but there are no experiments to show that brain octopamine as opposed to octopamine in the ventral nerve cord are responsible for the effects. Again, I am not suggesting that this already long manuscript needs more data, only that the nod to mechanism is more of statement of possibility than indication of genetic pathways, anatomic pathways, or neuronal circuits. The paper is more than just a "methods paper" because of the insights provided by this analysis of effects of environment on foraging, reproductive behavior and, potentially, decision making. The readers of *eLife* will like this one.

---

## [Author Response]

*[…] Essential revisions:*

*It is felt that the link to neural mechanism is rather a 'proof of principle', and does not go deep into the mechanism as may be possible with this model system. It was not felt that this is a critical flaw, but rather that the limitations need to be more clearly discussed, providing an opportunity to set the scene as to where research of this type could (and likely will) be directed in future.*

We agree that the analysis of the *Orco* and *Tβh* mutants is a first step towards a detailed mechanistic dissection of the behavioral adaptations happening upon changes in internal state. In the revised version we have modified the Introduction, the *Orco* manipulation section, the octopamine section and the Discussion to highlight this more clearly. We highlight the specific changes below and in the answers to the comments to each reviewer:

We have modified the second part of the last paragraph of the Introduction to highlight that the two mutation experiments *are examples* of the potential of our setup to dissect the underlying neuronal mechanisms of the behavior under study. We also removed the word “mechanistic” and described our results from the mutation experiments as “initial insights”:

“Importantly, we provide two examples on how our paradigm can be used in the dissection of the genetic and neuronal mechanisms underlying nutrient decisions: First, we show that olfaction is not required to reach protein homeostasis, but that it mediates the efficient recognition of yeast as an appropriate food source in mated females. Second, we show that octopamine mediates homeostatic postmating responses, but not the effects of internal sensing of amino acid deprivation state. Our study provides a quantitative description of how the fly changes behavioral decisions to achieve homeostatic nutrient balancing as well as initial insights into the mechanisms underlying protein homeostasis.”

The first paragraph of the *Orco* manipulation section now clearly states that this experiment is a proof of principle:

“As a proof of principle of how our setup could be used to uncover the neuronal mechanisms underlying foraging decisions, we decided to analyze the role of olfaction in nutrient homeostasis.”

To clarify that the results from the octopamine manipulation do not give us specific insights about the brain, we state that the *Tβh* manipulation affects the whole animal. We also added "to show that our setup could be used" to this sentence to highlight the fact that these experiments are more about exploring the usefulness of the tracking approach than an in depth exploration of molecular mechanisms:

“We therefore decided to show that our setup could be used to test possible neuromodulatory effects of octopamine on yeast foraging, using mutants for the gene encoding Tyramine β-hydroxylase (TβH), an enzyme required for the biosynthesis of octopamine in the whole animal.”

And in the last paragraph of the octopamine section, we highlight that this is just a “first step” in the dissection of the mechanisms:

“These results provide a first step towards dissecting the role of octopamine in nutrient homeostasis.”

We have added the word “behavioral” to emphasize that we are referring to the behavioral and not neuronal mechanisms: “Furthermore, these results demonstrate that one way in which mating increases yeast preference is by inducing long feeding bouts, allowing us to make first conclusions about the behavioral mechanisms behind changes in food choice”.

*Reviewer #1:*

*This paper addresses the very important challenge of identifying key drivers of nutritional decision-making using Drosophila as a tractable model organism. Furthermore, it employs advanced tracking to quantify the motion and precise feeding decisions of individuals among multiple resource patches.*

*This analysis shows that flies sense deficits in amino acids and can adjust their motion and feeding behavior to compensate, and that this is impacted by their mating state. Given that the experimental design was such as to remove (to a great degree) the effects of search, and rather to focus on the behavior of flies once on a patch, the analysis focused on the decisions to engage and to leave (as opposed to find) patches. The analysis includes highly detailed information as to how flies move on the patches.*

*An area which needs improving, in my mind, however, is at it relates to the neural mechanisms at work. This has the potential to be the most interesting part of the paper (and the one I was looking forward to getting to!), but in the end it provides useful, but indirect, evidence of the actual mechanisms at work. For example, the experiments suggest the importance of octopamine, but they do not get into the regions of the brain (or even if it is in the brain) important, and I think that while this section is helpful it is more of a start than a conclusion to the story, and perhaps the authors could make more clear what is, and what is not, shown by these data.*

In summary I find this a highly rigorous and useful paper.

We thank the reviewer for the positive evaluation of our work. We have modified the Introduction, the interpretation of the octopamine results and the Discussion to be clearer about what is and what is not shown by these data. Please see the above response to the Essential Revisions for details on the revised sections.

*Reviewer #2:*

*The authors have provided an extremely detailed behavioural analysis of nutrient-specific foraging behaviour in Drosophila, coupled with two experiments looking at the involvement of olfaction and octopamine in different stages of the control of protein homeostasis. Although the manuscript is extensive, the narrative is interesting and the illustrations beautifully composed. The paper adds significantly to the literature on appetite control and nutrient regulation. I have a few somewhat substantive issues.*

*Abstract: change "deeply" to "profoundly" or similar. I would also remove the second sentence, which is over-egging things and not strictly true.*

We have changed the word “deeply” as suggested and we have changed the second sentence of the Abstract to:

“A quantitative understanding of the behavioral changes upon metabolic challenges is key to a mechanistic dissection of how animals maintain nutritional homeostasis.”

*I felt the term "engaging the food" to be a little awkward (begs the question "Engaging food in what?"). "Stop at" is what the data indicate – or "arrested at the food". I pick up on the precision of language a little more below.*

We agree the word “engaging the food” might not be the best option. In our opinion what is important is to use a terminology that clearly differentiates between the animal stopping at the food and not moving at all (“resting”) and the animal stopping but being still active (e.g. “micromoving”). As "stop at" should be clear enough we have decided to follow your recommendation and use the term “stop at” the food patch instead of “engaging” the food patch. We have changed the manuscript and the figures accordingly.

*In the subsection “The lack of dietary AAs increases exploitation and local exploration of yeast patches”. This is more by way of a comment than a suggestion: the point is introduced and to some extent tested, but it is worth pointing out somewhere that the volume ingested during a feeding bout (meal) is a product of the duration of the feeding bout and the rate at which food is ingested across the meal. Meal size (and its volumetric and other determinants) has been shown in some insect systems to be positively correlated with ingestion rate and affected by common inputs (both homeostatic and non-homeostatic) (e.g. Animal Behav. 36: 1216-27). Hence, meal duration (which is estimated in the present set up) tends to be a more conserved variable than meal size (a fly can eat more by ingesting faster over the same meal duration). Changes in ingestion rate can result from greater amplitude of feeding central pattern generator (and hence motor) output, or greater frequency of rhythm, or through a change in intra-meal burst and interval structure. The data reported address and show a contribution from the latter, but changes in both amplitude and frequency of CPG output are also known.*

While food intake can theoretically be increased by an increase in the frequency of the feeding rhythm we had not found any evidence for such an effect in *Drosophila melanogaster* in a previous study (Itskov et al., Nature Communications, 2014). However, a careful analysis of the feeding rhythm data of the experiments in this manuscript revealed very small effects of internal states on sip durations and inter-sip-intervals. The effects are very small but we have decided to change the manuscript to include these data. In the two new panels D and E of Figure 4—figure supplement 3, we now analyze the effect of internal state on the feeding rhythm. We have added the following text to describe the observed effects:

“The volume ingested during a feeding bout is a product of the duration of that bout and the feeding rate. […] However, the mode of the sip duration distributions did not change when mated flies pre-fed a suboptimal diet were compared to females kept on a rich diet (0.12 s, *p* = 0.1196), but it decreased when flies were pre-fed the AA- diet (0.11 s, *p* = 0.0067).”

*In the fourth paragraph of the subsection “The lack of dietary AAs increases exploitation and local exploration of yeast patches” – 'intake' rather than 'use', which is a vague term. There are numerous other instances throughout the manuscript of faintly teleological terminology – "decision", "engage", "exploitation", "exploration", "choose to visit distant patches", etc. In such a beautiful analysis of behaviour as this, I would rather see clear, objective description. Maybe I'm being a pedant on this, but J.S. Kennedy (The New Anthropomorphism, 1993, CUP) was spot on in my view.*

We have done our best to carefully choose the words we used in this study. In general we tried to avoid any word which indicates that we are measuring food intake using video tracking or the flyPAD method. Although all our data indicate that both methods allow us to extrapolate food intake quite precisely, technically speaking we are not measuring food intake.

We have modified the text to minimize the use of the words mentioned by Dr. Simpson. We find however, that completely avoiding the use of words such as "decide" and "choose" is detrimental to the readability of the manuscript. Furthermore, these two terms are widely used in many studies describing animal behavior, including studies using invertebrates. See for example:

1) Bussell, Jennifer J., Nilay Yapici, Stephen X. Zhang, Barry J. Dickson, and Leslie B. Vosshall. 2014. “Abdominal-B Neurons Control *Drosophila* Virgin Female Receptivity.” Current Biology24 (14). In Abstract: "The female must parse her own reproductive state, the external environment, and male sensory cues to decide whether to copulate"

2) Palmer, Chris R., and William B. Kristan. 2011. “Contextual Modulation of Behavioral Choice.” Current Opinion in Neurobiology21 (4). "Both the nematode worm (*Caenorhabditis elegans*) [11] and the medicinal leech (Hirudo sp.) [12] decide whether to swim or to crawl based, in part, on the level of water they are in."

3) Barron, Andrew B, Kevin N Gurney, Lianne F S Meah, Eleni Vasilaki, and James A R Marshall. 2015. “Decision-Making and Action Selection in Insects: Inspiration from Vertebrate-Based Theories.” Frontiers in Behavioral Neuroscience9 (August). "The process of decision-making in the insect olfactory learning pathway is illustrated by how the system changes as an animal learns a specific odor is rewarding and changes its behavior to effect an appetitive response to the odor"

4) Dickson, Barry J. 2008. “Wired for Sex: The Neurobiology of *Drosophila* Mating Decisions.” Science322 (5903). "The female decides whether to accept or reject the male, depending on her perception of his pheromone and acoustic signals, as well as her own readiness to mate."

5) Haberkern, Hannah, and Vivek Jayaraman. 2016. “Studying Small Brains to Understand the Building Blocks of Cognition.” Current Opinion in Neurobiology37. "The strongest behavioral evidence that insects must use abstract internal representations comes from honeybees, which display behaviors akin to deliberative decision-making."

6) Yang, Chung-Hui, Priyanka Belawat, Ernst Hafen, Lily Y Jan, and Yuh-Nung Jan. 2008. “*Drosophila* Egg-Laying Site Selection as a System to Study Simple Decision-Making Processes.” Journal Article. Science319 (5870). "The first indications that egg-laying site selection may employ a simple decision-based process emerged from our observations of females as they lay eggs."

We have therefore felt it best to not completely remove these two words from our manuscript.

*The one counter-intuitive finding is that reported in the subsection “Amino acid challenges reduce global exploration and increase revisits to the same yeast patch”. My understanding is that theory and experiment (Levy flight; intrinsic (local) search vs. extrinsic search (ranging); Dethier's fly dance, etc.) predicts that flies should stay nearby "high quality" patches and roam more extensively when on "lower quality" patches. Staying nearby to an imbalanced food source when in a state of AA imbalance and roaming globally when AA balanced appears contrary to expectations and suggests something interesting, perhaps about the evolved predicted spatial distribution of protein resources in the environment, e.g. complementary imbalanced AA sources are likely to be close by to one another. In the last paragraph of the aforementioned subsection there is an unconvincing statement on this issue, which needs some more thought I think.*

Unfortunately, we think that the point raised by Dr. Simpson stems from a misunderstanding of our experimental design. We recognize that we have to be clearer in our description. The AA+ rich (balanced) and AA+ suboptimal (imbalanced) labels refer to the dietary pre-treatment of the flies 3 days prior to the foraging assay. During the assay, all flies irrespective of their internal state were tested in an arena containing food patches of the same quality (9 food patches with yeast at 180g/L and the other 9 with sucrose at 180g/L). This is important as the experimental design aims at describing the effect of different internal state on foraging in the same environment. Therefore, since we don’t test the behavior of flies around patches of varying qualities, our results don’t contradict the previously reported findings. We therefore consider the statement valid: “Taken together, these changes in exploratory strategy should enable the fly to efficiently increase the intake of yeast while minimizing the distance travelled to the next spot.”

Because we consider this is a crucial point that should be very clear in our manuscript, we have now modified both the text and the figures to clarify the experimental design and our findings more clearly. Specifically:

When describing the diets used for the metabolic state manipulations, we added a sentence that clarifies that these diets were given before the assay: “we manipulated the metabolic state of the flies by letting them feed ad libitum on a chemically defined (holidic) medium (Piper et al., 2014) during three days prior to the foraging assay. This holidic medium allows us to specifically manipulate amino acids (AA) in the diet”.

In Figure 2, the holidic medium label now says: “Dietary pre-treatment (holidic medium):”

We changed the text in the last paragraph of the Introduction: "Furthermore, we describe how the internal deficit of dietary amino acids increases exploitation of proteinaceous patches and restricts global exploration and how these behaviors dynamically shift towards increasing exploration as the fly reaches satiation.”

In Figure 5 we have added the label “Dietary pre-treatment” on top of the text describing the internal state conditions and we have added the labels “Yeast 180 g/L” and “Sucrose 180 g/L” on top of the representative trajectories in panels A, B and C. Furthermore, in all figures containing panels with box plots, we have changed the label referring to the metabolic state from “AA” to “AA pre-treatment”.

At the beginning of the Results section where we describe the tracking setup, we explain: “To capture how flies decide what food to eat, we built an automated image-based tracking setup (Figure 1) that captures the position of a single *Drosophila melanogaster* in a foraging arena (Figure 1) containing 9 yeast patches (amino acid source) and 9 sucrose patches (carbohydrate source) of equal concentration.”

Substrate labels in Figure 1 now read: “Yeast 180g/L” and “Sucrose 180 g/L”.

We hope that with these changes clarify the experimental design.

*Subsection “ORs mediate efficient recognition of yeast as an appropriate food source”. This section could be set up by referencing back to the observation reported earlier that arrival rates at food sources was not impacted by state – implying that olfactory cues acting over a distance are not involved, but close-range chemosensory cues (short-range olfactory and gustatory) are more likely. Would help build the narrative.*

We thank Dr. Simpson for this suggestion. We had considered using this narrative but decided not to do so given that the encounter rate increases in AA-deprived flies.

*The Discussion is where you need to address the matter mentioned above in the subsection “Amino acid challenges reduce global exploration and increase revisits to the same yeast patch”. The blowfly dance results seem at odds with present findings – flies dance more vigorously and stay longer locally after stimulation with a higher concentration food source. The same is the case for bees and other animals (William Bell's work – e.g. see his 1990 book). It is true that greater levels of deprivation elicit greater local dancing to a given food; but still the effect is strongly dependent on food quality relative to state.*

We hope that we have clarified this issue in the preceding answer to your earlier comments. As we think that the interpretation is still correct and there is no conflict with previous blowfly dance results we decided not to modify the text in the subsection “Amino acid challenges reduce global exploration and increase revisits to the same yeast patch”.

*Reviewer #3:*

*Ribeiro's group uses this paper to establish a new assay to assess the relationships between internal state defined by the interaction between sexual experience (mated or virgin) and metabolic experience (more or less amino acids in the medium) on behaviour (explore or exploit). I like the analysis very much and I think it will be a welcome addition to the literature. Their method includes potential to dissect decision making at the level of behavior and offers a world of new possibilities for evaluating the contribution of genes, development, and physiology to behavior. They note and promise that they are providing insight into mechanism as well and this part of the paper pales when compared to the behavioral analysis. The nod to mechanism occurs via the manipulation of Orco to demonstrate a role for smell in the behavioural strategy used by mated females. Not clear that it plays a role for the virgins because there are no data and it seems less likely but entirely possible that olfaction is used to detect or evaluate sugars. I do not think more experiments need to be performed but I think the Orco data provide more of a proof of principle than an exposition of mechanism and the paper would be improved if these data were presented that way. This same point applies to the tyramine β hydroxylase experiments. The paper is written as if these data provide insight into the brain, but there are no experiments to show that brain octopamine as opposed to octopamine in the ventral nerve cord are responsible for the effects. Again, I am not suggesting that this already long manuscript needs more data, only that the nod to mechanism is more of statement of possibility than indication of genetic pathways, anatomic pathways, or neuronal circuits. The paper is more than just a "methods paper" because of the insights provided by this analysis of effects of environment on foraging, reproductive behavior and, potentially, decision making. The readers of eLife will like this one.*

We appreciate the positive assessment of our work and thank the reviewer for the pertinent remarks. We have modified several sections of the manuscript to present the *tyramine β hydroxylase* manipulation results and the *Orco* manipulation results as proof of principle rather than mechanistic insights into neuronal processes (see also answers to major comments of Reviewer #1). In summary:

In the last paragraph of the Introduction we highlight that the two mutation experiments *are examples* of the potential of our setup to dissect neuronal mechanisms of the behavior under study. We also removed the word “mechanistic” and describe the results from the experiments with the mutants as “initial insights”.

We introduce the *Orco* manipulation as a proof of principle experiment (subsection “ORs mediate efficient recognition of yeast as an appropriate food source”, first paragraph).

In the octopamine section we added "we therefore decided to show that our setup could be used to test possible neuromodulatory effects of octopamine on yeast foraging" to this sentence to highlight the fact that these experiments are more about exploring the usefulness of the tracking approach than an in depth exploration of molecular mechanisms. And in the conclusion of this section, we highlight that this is just a “first step” in the dissection of the mechanisms.

In the Discussion to remove any association of our results with specific insights about the brain (second paragraph), we changed the word “brain” by “nervous system” and highlight that it is the accumulated evidence (not just our results) which lead us to conclude that the nervous system might use different mechanisms to change specific aspects of behavioral outcomes.

Furthermore, we have modified the Abstract to clearly state that the results of the *Orco* manipulation apply only to mated females:

“Finally, we show that olfaction mediates the efficient recognition of yeast as an appropriate protein source in mated females”.

We have also added this clarification in the conclusion of the results of the *Orco* manipulation section:

“Taken together these results show that, in mated females, OR-mediated olfaction is necessary for efficient recognition of yeast as an appropriate resource but is not required to locate food patches at a short range or to achieve nutritional homeostasis.”

Finally, we have also included the term “mated” in the Discussion: “Our data suggest that while olfactory-impaired matedflies…”

Regarding the detection of sugars, we are aware that some few studies have suggested that flies can detect sugar at a distance. While this is therefore a possibility we have done our best to minimize that our sucrose patches emit volatiles. We use sucrose with 99.5% purity mixed with ultrapure agarose and milliQ water to prepare our sucrose patch solutions. The sugar patches in our foraging arena should therefore not have any volatiles that can be detected by the flies using olfaction. As the experimental evidences for olfactory detection of sugars at a distance are scarce and we minimize the probability that our sucrose patches emit volatiles we think it is unlikely that flies use olfaction to evaluate sugars. However, as always in science one never knows.